# Single molecule analysis of CENP-A chromatin by high-speed atomic force microscopy

**Daniël P Melters[1]\*, Keir C Neuman[2]\*, Reda S Bentahar[1], Tatini Rakshit[1,3], Yamini Dalal[1]\***

[1]National Cancer Institute, Center for Cancer Research, Laboratory Receptor Biology and Gene Expression, Bethesda, United States; [2]National Heart, Lung, and Blood Institute, Laboratory of Single Molecule Biophysics, Bethesda, United States; [3]Department of Chemistry, Shiv Nadar University, Dadri, India

**Abstract** Chromatin accessibility is modulated in a variety of ways to create open and closed chromatin states, both of which are critical for eukaryotic gene regulation. At the single molecule level, how accessibility is regulated of the chromatin fiber composed of canonical or variant nucleosomes is a fundamental question in the field. Here, we developed a single-molecule tracking method where we could analyze thousands of canonical H3 and centromeric variant nucleosomes imaged by high-speed atomic force microscopy. This approach allowed us to investigate how changes in nucleosome dynamics in vitro inform us about transcriptional potential in vivo. By high-speed atomic force microscopy, we tracked chromatin dynamics in real time and determined the mean square displacement and diffusion constant for the variant centromeric CENP-A nucleosome. Furthermore, we found that an essential kinetochore protein CENP-C reduces the diffusion constant and mobility of centromeric nucleosomes along the chromatin fiber. We subsequently interrogated how CENP-C modulates CENP-A chromatin dynamics in vivo. Overexpressing CENP-C resulted in reduced centromeric transcription and impaired loading of new CENP-A molecules. From these data, we speculate that factors altering nucleosome mobility in vitro, also correspondingly alter transcription in vivo. Subsequently, we propose a model in which variant nucleosomes encode their own diffusion kinetics and mobility, and where binding partners can suppress or enhance nucleosome mobility.

**\*For correspondence:**
daniel.melters@nih.gov (DPM);
neumankc@nhlbi.nih.gov (KCN);
dalaly@mail.nih.gov (YD)

## Editor's evaluation

This is an interesting paper that describes the validation of high speed AFM as a tool for measuring the dynamics of individual nucleosomes in vitro. After validating the methodology the authors go on to apply the method to nucleosomes containing the centromere-specific histone variant CENP-A, and they show that addition of the CENP-A binding factor CENP-C radically alters the mobility of centromeric nucleosomes.

## Introduction

Regulating physical access to DNA is central to both gene expression, genome topology, and genome integrity across eukaryotes. Decades of data strongly suggest that more physically accessible chromatin is also more transcriptionally permissive (*Klemm et al., 2019*; *Maeshima et al., 2019* landmark papers from *Weintraub and Groudine, 1976*; *Wu et al., 1979*). In contrast, compacted chromatin or heterochromatin has been correlated with transcriptional restriction (*Allshire and Madhani, 2018*; *Flamm et al., 1969*; *Janssen et al., 2018*; *Schultz, 1936*). These chromatin states are dynamic and

subject to tight regulation. Chromatin-binding proteins dictate many of these dynamics in part driven by the presence and deposition of specific post-translational modifications (PTMs) of nucleosomes (*Chung et al., 2023*; *Rothbart and Strahl, 2014*; *Taverna et al., 2007*; *Tolsma and Hansen, 2019*). For instance, HP1 binds to H3K9me2/3 nucleosomes (*Bannister et al., 2001*; *Nakayama et al., 2001*; *Lachner et al., 2001*; *Sanulli et al., 2019*), ultimately resulting in transcriptionally repressive chromatin (*Bannister et al., 2001*; *Escobar et al., 2021*; *Hwang et al., 2001*; *Nakayama et al., 2001*; *Lachner et al., 2001*). Interestingly, upon binding of HP1 to H3K9me3 nucleosomes, these nucleosomes' internal residues become more exposed to hydrogen/deuterium exchange (*Sanulli et al., 2019*). This seminal finding suggests that chromatin-binding proteins not only serve as a recruitment platform for other binding proteins but can rapidly alter the physical properties of individual nucleosomes. The logical extension of this concept is examining how altering innate physical properties of nucleosomes modulates the local chromatin state's structure and function.

To study how nucleosome dynamics is altered by chromatin binding factors, single molecule techniques have been developed, ranging from groundbreaking in vitro techniques such as optical and magnetic tweezers (*Bustamante et al., 2021*; *Chien and van Noort, 2009*; *Killian et al., 2018*; *Neuman and Nagy, 2008*) and in vivo single molecule tracking (*Iida et al., 2022*; *Izeddin et al., 2014*; *Mueller et al., 2013*; *Shen et al., 2017*; *van Staalduinen et al., 2023*; *Wang et al., 2021*). The former two techniques rely on precisely designed DNA sequences and constructs to guarantee precise measurements. By manipulating an optical trap or magnetic tweezer, tension and torsion forces can be exerted on the associated DNA molecule, which in turn alters the forces exerted on nucleosomes (*Bustamante et al., 2021*; *Chien and van Noort, 2009*; *Killian et al., 2018*; *Neuman and Nagy, 2008*). In contrast, single molecule tracking in cells is made possible by photostable fluorophores covalently bound to a target protein. These tagged proteins are introduced into cells at low concentration to allow the tracking of single molecules, with limited control where the tagged proteins will go (*Iida et al., 2022*; *Mueller et al., 2013*; *Shen et al., 2017*; *Wang et al., 2021*). Both systems are powerful and have distinct advantages, ranging from bp-precision of nucleosome sliding, folding-unfolding dynamics to determining residency time of transcription factors. However, a gap exists connecting these two technical approaches, namely assessing, and quantifying the dynamics of individual nucleosomes and correlating them with global chromatin dynamics. High-speed atomic force microscopy (HS-AFM) has the capability to span this gap. It is an in vitro based technique that permits real time tracking of single molecules in the context of interacting macromolecular complexes (e.g. myosin tracking on actin filaments, *Ando, 2018*; chromatin techniques reviewed in *Melters and Dalal, 2021*). By imaging nucleosome arrays in buffer over time, it is possible not just to track, but also quantify the motions of individual nucleosomes within an array.

In addition to PTMs, the chromatin landscape is also marked by the local enrichment of histone variants (*Buschbeck and Hake, 2017*; *Jamge et al., 2023*; *Martire and Banaszynski, 2020*; *Melters et al., 2015*), such as the centromere-specific H3 histone variant CENP-A/CENH3. CENP-A nucleosomes recruit several centromeric proteins (*Walstein et al., 2021*; *Mendiburo et al., 2011*; *Régnier et al., 2005*), including CENP-C. CENP-C in turn functions as the blueprint for the formation of the kinetochore (*Cheeseman et al., 2006*; *DeLuca and Musacchio, 2012*; *Hara et al., 2023*; *Walstein et al., 2021*; *Przewloka et al., 2007*; *Weir et al., 2016*; *Yatskevich et al., 2023*). Recently, we reported that the central domain of CENP-C (CENP-C$^{CD}$) induces loss of CENP-A nucleosomal elasticity in silico and in vitro (*Melters et al., 2019*). This finding correlates with decreased hydrogen/deuterium exchange of CENP-A nucleosomes when bound by CENP-C$^{CD}$ (*Falk et al., 2016*; *Falk et al., 2015*; *Guo et al., 2017*). Interestingly, CENP-C knock-down resulted in downregulation of centromeric transcription (*Bury et al., 2020*), whereas CENP-C overexpression resulted in reduced RNA polymerase 2 (RNAP2) levels at the centromere and clustering of centromeric chromatin (*Melters et al., 2019*). We were curious about how CENP-A alone, and in combination with CENP-C mechanistically impacts centromeric chromatin fiber mobility. To address this question, we employed HS-AFM to track thousands of canonical or centromeric nucleosomes in arrays in real time. We report that HS-AFM imaging is free of tip-induced artifacts and CENP-A chromatin responds predictably to various control conditions. Next, we find that the essential kinetochore protein CENP-C, which is CENP-A chromatin's closest binding partner, directly impacts nucleosome mobility and, surprisingly, also chromatin fiber motion in vitro. These data represent a technological advance in imaging and analyzing chromatin dynamics by HS-AFM. We extended these findings in vivo using immunofluorescence imaging and biochemical

approaches, reporting that overexpressing CENP-C alters centromeric chromatin transcription and the ability to load new CENP-A molecules. Cumulatively, these data support the notion that local transcriptional competency depends on innate properties and local homeostasis of histone variants within the chromatin fiber in vivo.

## Results

We are interested in understanding how nucleosomes 'behave' in biologically relevant conditions. Elegant single-molecule techniques have enabled us to understand details about the movement of transcription factors inside the nucleus (*Iida et al., 2022*; *Mueller et al., 2013*; *Shen et al., 2017*; *Wang et al., 2021*) and how torsion and pulling/pushing forces influence nucleosomes (*Bustamante et al., 2021*; *Chien and van Noort, 2009*; *Killian et al., 2018*; *Neuman and Nagy, 2008*). Although the behavior of a single trajectory might be stochastic, the statistical behavior from many trajectories may reveal additional physical properties, such as diffusion and folding-unfolding dynamics.

By directly observing topographic characteristics and dynamics of chromatin using HS-AFM, we were able to assess the motions of individual nucleosomes in real-time (*Figure 1*). This emerging single-molecule technique is powerful, and shares similarities with live cell imaging (*Ashwin et al., 2019*; *Specht et al., 2017*), magnetic tweezers, and optical tweezers (*Bustamante et al., 2021*; *Chien and van Noort, 2009*; *Killian et al., 2018*; *Neuman and Nagy, 2008*). Whereas, live cell imaging and single-molecule force spectroscopy methods rely on fluorophore-tags and tethering, HS-AFM can be done on both unmodified and modified protein all while requiring minimal sample preparation (*Ando, 2018*).

Recently, we showed by HS-AFM that, at a qualitative level, H3 nucleosomes are mobile and make intermittent contact with other H3 nucleosomes (*Melters and Dalal, 2021*). Here, we set out to quantify the dynamics of individual nucleosomes in the context of chromatin. This means that we observed and quantified global nucleosome movement on mica surface, which can be one-dimenstional motion of nucleosome sliding along DNA events and two-dimensional whole chromatin fiber movements (*Figure 1*). Nucleosomes that move away from the mica surface into the buffer solution could not be tracked. We predict that these global nucleosome motions reflect the complex dynamics nucleosomes display in the nucleus (*Ide et al., 2022*). To assay CENP-A chromatin, we reconstituted CENP-A nucleosomes on a~3.5 kbp plasmid containing four copies of α-satellite repeats or a~8.5 kbp plasmid containing three copies of the PCAT2 lncRNA gene. After verifying successful in vitro chromatin reconstitution (*Figure 1—figure supplement 1*, *Figure 1—figure supplement 1—source data 1 and 2*), we imaged CENP-A chromatin using the Cypher VRS system (see Materials and methods for details). Videos were obtained at one second per frame. When the AFM tip scans the mica surface, it first goes from the top to the bottom, followed by from the bottom to the top. To avoid discontinuous scanning of the same regions of the mica surface, we limited our analysis to every other frame, resulting in an effective scanning rate of two seconds per frame. Next, the videos were converted to TIFF sequences. Using MATLAB's single particle tracking MatlabTrack package, we obtained a total of 6052 individual nucleosome trajectories from 13 different samples (see Materials and methods for details). To obtain significant mean square displacement (MSD) values, individual trajectories needed to be of sufficient duration. Therefore, we selected individual trajectories that were at least 20 s, with a maximum of 120 s (10–60 points). In addition, we wanted to make sure that we were tracking the same nucleosomes, so we estimated the maximum single step size at twice the nucleosome width or 24 nm. Furthermore, we were also careful about potential nucleosomes sticking to the mica surface, which led us to reject trajectories with an average R-step size (the displacement between two successive images) smaller than 1 nm and a maximum R range smaller than 8 nm. Next, we corrected each video for drift (*Figure 1—figure supplement 2*). Finally, we analyzed 3835 trajectories to obtain the step angle, step size, mean square displacement (MSD), and diffusion constant of individual nucleosomes (*Figure 1*).

### HS-AFM imaging is free of tip-induced artifacts

AFM is a topographical imaging technique that creates a sub-nanometer scale topological map of biological samples. A common concern is that the AFM tip may alter the sample during scanning, as the AFM tip moves in a zig-zag manner across the sample, potentially moving biological material

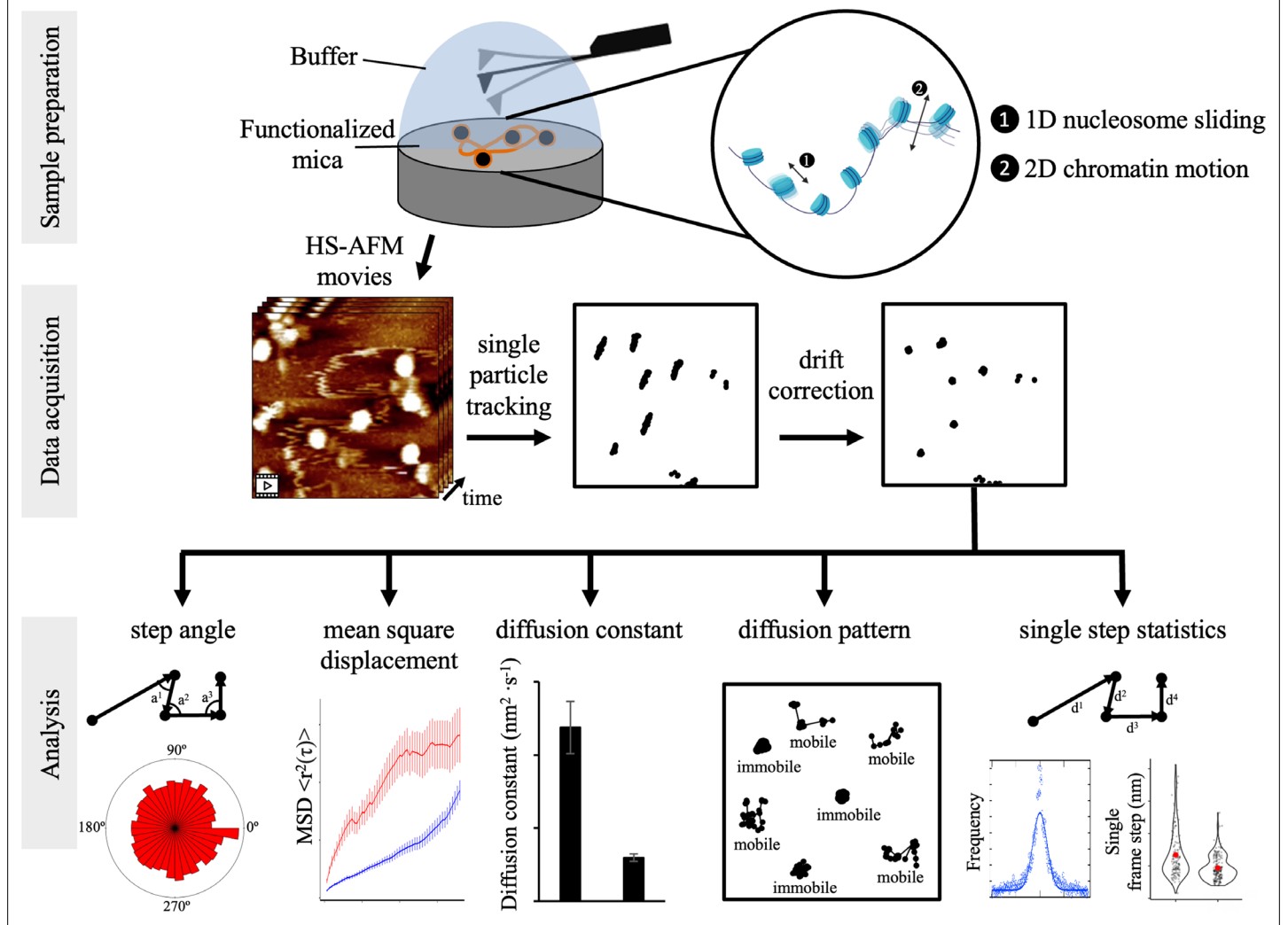

**Figure 1.** Schematic of experimental configurations for HS-AFM nucleosome measurements and analysis of extracted single particle trajectories Sample preparation: CENP-A nucleosomes were in vitro reconstituted and imaged by HS-AFM in fluid. By HS-AFM, we can track nucleosome motion, corresponding to both nucleosomes sliding along the DNA and chromatin fiber moving. Data acquisition: HS-AFM videos were obtained at a framerate of 0.5 Hz (2 s per frame) for a minimum of 20 s and up to 120 s. Using MATLAB, we extracted nucleosome trajectories, which were subsequently corrected for drift. Analysis: trajectories were analyzed to extract several mobility and diffusion-related parameters to determine both potential tip-scanning artifacts and to characterize nucleosome dynamics.

The online version of this article includes the following source data and figure supplement(s) for figure 1:

**Figure supplement 1.** Confirmation of *in vitro* reconstitution of CENP-A chromatin.

**Figure supplement 1—source data 1.** Complete gels for *Figure 1—figure supplement 1A* (12% SDS-PAGE) and 1 C (1% agarose gel).

**Figure supplement 1—source data 2.** Nucleosome statistics for each nucleosome for nucleosome diameter (nm), nucleosome height (nm), and nucleosome volume (nm³).

**Figure supplement 2.** Drift correction of HS-AFM videos.

in its path. To determine whether there is indeed a tip effect, we performed several controls. We reasoned that if the AFM tip altered samples during scanning, it would create a distinctive signature in single particle trajectories. We imaged CENP-A chromatin under multiple conditions (*Figure 2— videos 1–6*). First, we assessed the angle between successive steps of the nucleosomes (*Figure 2A*). If there was tip-induced drift, there would be a bias in the distribution of angles. We did not observe a bias in the angle distributions (*Figure 2B*; *Figure 2—figure supplement 1A*, *Figure 2—source data 1*). Second, we looked at the diffusion constants over the x-axis alone or the y-axis alone (*Figure 2C*) in the absence or presence of 1-(3-aminopropyl) silane (APS). APS functionalizes the mica surface

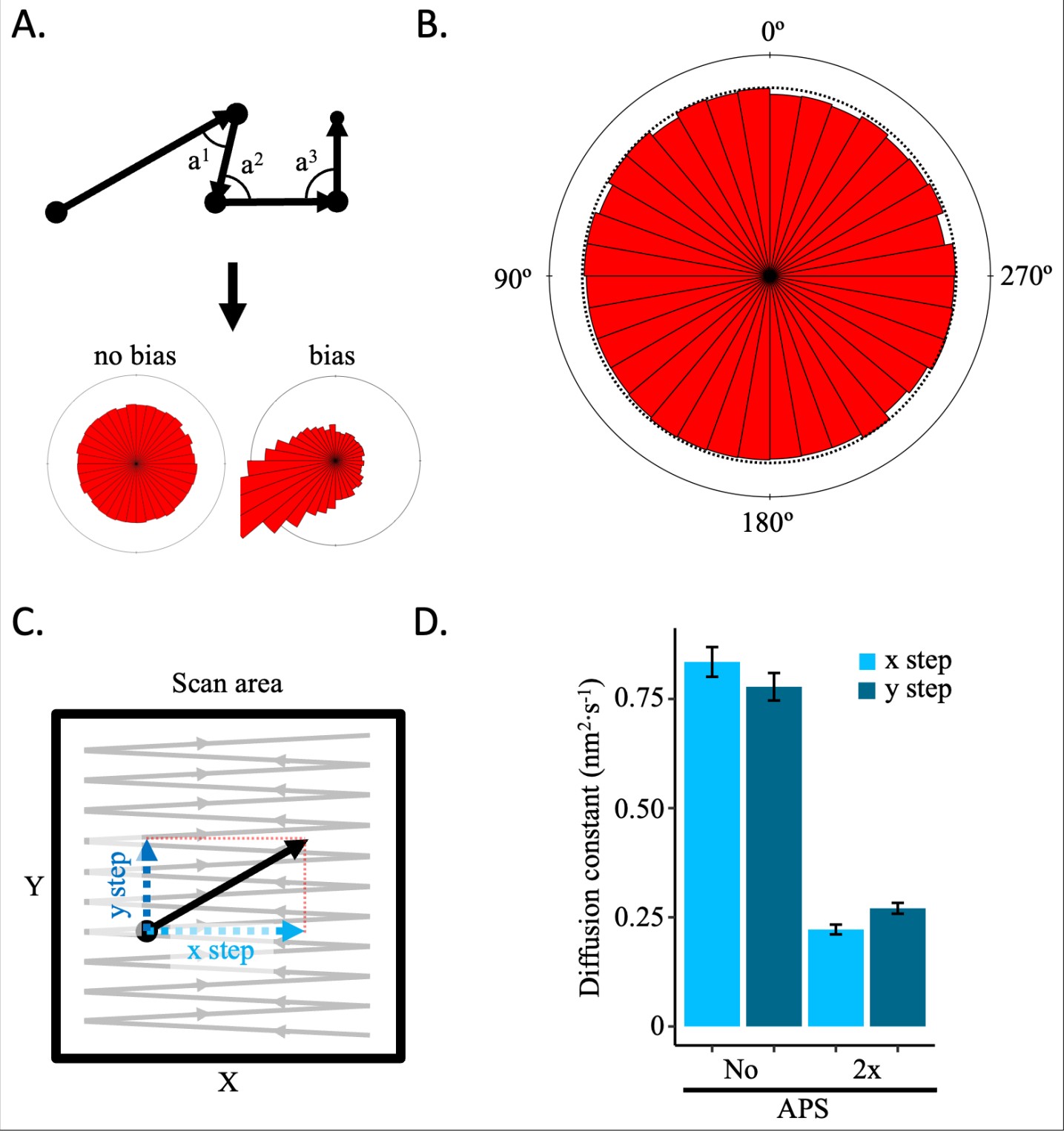

**Figure 2.** No AFM tip-motion effect was observed. If the AFM tip were to displace the sample during scanning, it should result in a motion bias in the direction of scanning that can be detected. (**A**) Schematic representation of how angle between successive steps within a trajectory is determined and representative angle distribution graphs for no bias or bias. (**B**) All angles for successive steps of all trajectories of five control conditions (low salt, high salt, no APS, 2 x APS, and Tween-20) show no sign of bias (*Figure 2—figure supplement 1A*). (**C**) Every step has an x and y coordinate. By obtaining the diffusion constant for each axis separately, motion bias between the x and y axis can be discerned, which is an indication of bias introduced by the AFM tip. (**D**) The diffusion constants for the x and y axis for CENP-A nucleosomes in low or high salt conditions show differences between imaging conditions, but not within each condition (*Figure 2—figure supplement 1B*). The error bars represent the standard error.

*Figure 2 continued on next page*

*Figure 2 continued*

The online version of this article includes the following video, source data, and figure supplement(s) for figure 2:

**Source data 1.** Trajectory statistics for all tracked nucleosomes.

**Source data 2.** Basic statistics for all tracked nucleosomes.

**Figure supplement 1.** No AFM tip-motion effect observed across control conditions.

**Figure supplement 1—source data 1.** Trajectory statistics for all tracked nucleosomes.

X and Y coordinates, distance between two steps, accumulated distance per trajectory, and angle between two steps are reported.

**Figure supplement 2.** The scatter plot of average step size over each frame of the videos shows no bias by the AFM tip, as the $R^2$ linear regression is 0.15.

**Figure supplement 2—source data 1.** Trajectory statistics for all tracked nucleosomes.

X and Y coordinates, distance between two steps, accumulated distance per trajectory, and angle between two steps are reported.

**Figure supplement 2—source data 2.** Basic statistics for all tracked nucleosomes.

**Figure 2—video 1.** HS-AFM video of in vitro reconstituted CENP-A chromatin in 5 mM NaCl containing buffer (low salt; speed =2 x).
https://elifesciences.org/articles/86709/figures#fig2video1

**Figure 2—video 2.** HS-AFM video of in vitro reconstituted CENP-A chromatin in 150 mM NaCl containing buffer (high salt; speed =2 x).
https://elifesciences.org/articles/86709/figures#fig2video2

**Figure 2—video 3.** HS-AFM video of in vitro reconstituted CENP-A chromatin imaged without functionalized mica (no APS; speed =2 x).
https://elifesciences.org/articles/86709/figures#fig2video3

**Figure 2—video 4.** HS-AFM video of in vitro reconstituted CENP-A chromatin imaged with double the amount of APS to functionalize mica (2 x APS; speed =2 x).
https://elifesciences.org/articles/86709/figures#fig2video4

**Figure 2—video 5.** HS-AFM video of in vitro reconstituted CENP-A chromatin imaged with physiological buffer +0.01% Tween (Tween; speed =2 x).
https://elifesciences.org/articles/86709/figures#fig2video5

**Figure 2—video 6.** HS-AFM video of in vitro reconstituted CENP-A chromatin on PCAT2 plasmid imaged with physiological buffer (DNA; speed =2 x).
https://elifesciences.org/articles/86709/figures#fig2video6

with positively charged amino groups, binding nucleic acid molecules under physiological conditions, allowing for the ability to image in air, in fluid, as well as perform force spectroscopy (*Lyubchenko et al., 2014*; *McAllister et al., 2005*; *Melters et al., 2019*; *Melters and Dalal, 2021*; *Rakshit et al., 2020*; *Shlyakhtenko et al., 2003*). We predicted that adding an excess of APS (333 nM APS is 2 x APS) would lower the diffusion constant compared to not adding APS (no APS). Indeed, we measured a lower diffusion constant for 2 x APS than no APS (*Figure 2D*, *Figure 2—source data 1 and 2*). If there were a tip effect, we reasoned there would be a difference in the diffusion constants between the x-axis and y-axis, which are parallel and perpendicular to the direction of tip scanning, respectively. We did not observe a bias in the diffusion constants between the two axes (*Figure 2D*; *Figure 2—figure supplement 1B*, *Figure 2—source data 1 and 2*). Third, to test for the potential impact of prolonged imaging on manipulation of trajectories, we assessed the step size distribution over time. We did not observe a significant effect of the video length on the step size (*Figure 2—figure supplement 2*, *Figure 2—source data 1 and 2*), indicating that extended imaging did not alter the trajectory dynamics. We are therefore confident that HS-AFM does not introduce a tip effect on chromatin and that drift is adequately corrected.

## Salt and APS concentrations impact chromatin dynamics in an anticipated manner

Next, we set out to determine how different buffer conditions impact CENP-A chromatin dynamics. Different salt concentrations are known to impact chromatin compaction and dynamics (*Allahverdi et al., 2015*; *Brasch et al., 1971*; *Yager et al., 1989*; *Yager and van Holde, 1984*). At lower salt concentrations (below 50 mM NaCl), nucleosomes are stabilized, whereas at higher salt concentrations (above 100 mM NaCl) nucleosomes become unstable. Here, we tested the effect of low salt concentrations (5 mM NaCl) versus high salt concentration (150 mM NaCl) on nucleosome dynamics by HS-AFM. For each condition we tracked 124 and 161 nucleosome trajectories, respectively (*Table 1*, *Figure 3—videos 1 and 2*). As expected, the MSD curve for high salt had a larger slope than for low

**Table 1.** Quantifications of HS-AFM videos.

Either CENP-A or H3 nucleosomes were in vitro reconstituted on plasmid DNA and imaged in fluid in the presence or absence of either 2.2-fold excess CENP-C$^{CD}$ or 0.2-fold excess of H1.5. $n$, number of nucleosome trajectories tracked. For each condition, at least three independent replicates were performed.

| Sample | $n$ | Number of steps | Average Diffusion constant (nm$^2$/s) | Average step size (nm) | Maximum R-step (nm) | R-step range (nm$^2$) |
|---|---|---|---|---|---|---|
| CENP-A nucleosomes | 498 | 13,989 | 2.3±0.2 | 4.2±0.1 | 11.2±0.2 | 23.2±0.6 |
| +1 x CENP-C$^{CD}$ | 368 | 7790 | 2.1±0.1 | 3.1±0.2 | 12.6±0.4 | 23.7±0.7 |
| +2 x CENP-C$^{CD}$ | 310 | 9063 | 0.78±0.06 | 2.8±0.1 | 8.0±0.2 | 14.3±0.4 |
| +4 x CENP-C$^{CD}$ | 166 | 7034 | 0.61±0.05 | 2.3±0.2 | 8.6±0.4 | 15.4±0.7 |
| CENP-A controls | | | | | | |
| Low salt | 124 | 3783 | 1.2±0.2 | 2.8±0.1 | 9.6±0.4 | 18.3±0.9 |
| High salt | 161 | 4887 | 4.1±0.3 | 5.5±0.2 | 14.6±0.4 | 40±2 |
| No APS | 244 | 5186 | 7.5±0.5 | 6.0±2.0 | 15.9±0.4 | 43±2 |
| 2 x APS | 120 | 2520 | 2.5±0.3 | 3.8±0.2 | 11.4±0.5 | 21±1 |
| Tween-20 | 587 | 16,126 | 5.9±0.2 | 6.0±0.1 | 14.7±0.2 | 43±1 |
| PCAT2 DNA | 710 | 15,497 | 5.7±0.4 | 5.8±0.2 | 16.0±0.2 | 33.7±0.6 |
| H3 controls | | | | | | |
| H3 nucleosomes | 66 | 1109 | 2.5±0.3 | 4.2±0.2 | 9.9±0.5 | 19±1 |
| +H1.5 | 391 | 8492 | 3.2±0.3 | 4.7±0.1 | 12.3±0.3 | 26.9±0.8 |
| H3 mononucleosome | 90 | 1344 | 9.3±0.9 | 7.9±0.3 | 17.4±0.5 | 47±2 |

salt (*Figure 3A*, *Figure 3—figure supplement 1A*, *Figure 3—source data 1 and 2*). This is reflected in the diffusion constant, which was 1.2±0.2 nm$^2$·s$^{-1}$ for low salt and 4.1±0.3 nm$^2$·s$^{-1}$ for high salt (*Figure 3B*, *Figure 3—figure supplement 1B*, *Figure 3—source data 1 and 2*).

To better understand the step dynamics, we first assessed the average single-frame step size and found that the average step size of CENP-A nucleosomes for high salt is double the length of that of low salt (5.5±0.2 nm vs 2.8±0.1 nm, respectively, *Table 1*, *Figure 3—figure supplement 1C*, *Figure 3—source data 2*). Furthermore, we calculated the R-step. The R-step is the single-frame displacement in the plane of the trajectory and is defined as the square root of the sum of the squares of the displacement in the x and y directions. The maximum R-step was only slightly different between the low and high salt conditions (9.6±0.4 nm vs 14.5±0.4 nm, respectively, *Table 1*, *Figure 3—figure supplement 1D*, *Figure 3—source data 2*). When we looked at the total range of a trajectory by calculating the R-step range, we observed a much larger variance in the distribution for the high salt compared to the low salt (40±2 nm$^2$ vs 18.3±0.9 nm$^2$, respectively, *Table 1*, *Figure 2—figure supplement 1E*, *Figure 3—source data 2*). These results are in line with how salt concentration is known to impact chromatin dynamics (*Allahverdi et al., 2015*; *Brasch et al., 1971*; *Yager et al., 1989*; *Yager and van Holde, 1984*).

Next, we plotted the distributions of single step displacements in the x-axis and y-axis and fitted them to a single Gaussian distribution, which is expected for a diffusive process. Although the x- and y- single-step distributions were poorly fit by single Gaussian distributions (D$_0$), they were well-fit by a sum of two Gaussian distributions (D$_1$ and D$_2$) (*Figure 3D*, *Figure 3—figure supplements 2 and 3*, *Figure 3—source data 1 and 2*). The relative fraction of the two Gaussian distributions differed between low and high salt conditions (*Figure 3E*, *Figure 3—figure supplements 2 and 3*, *Figure 3—source data 1 and 2*). These data indicate that there are two distinct populations of nucleosomes in both low- and high-salt conditions. The D$_1$ Gaussian population represents slower diffusing nucleosomes, potentially reflecting transient pausing, whereas the D$_2$ Gaussian population represents faster diffusing nucleosomes. As AFM can only detect samples that are on the mica surface, one concern

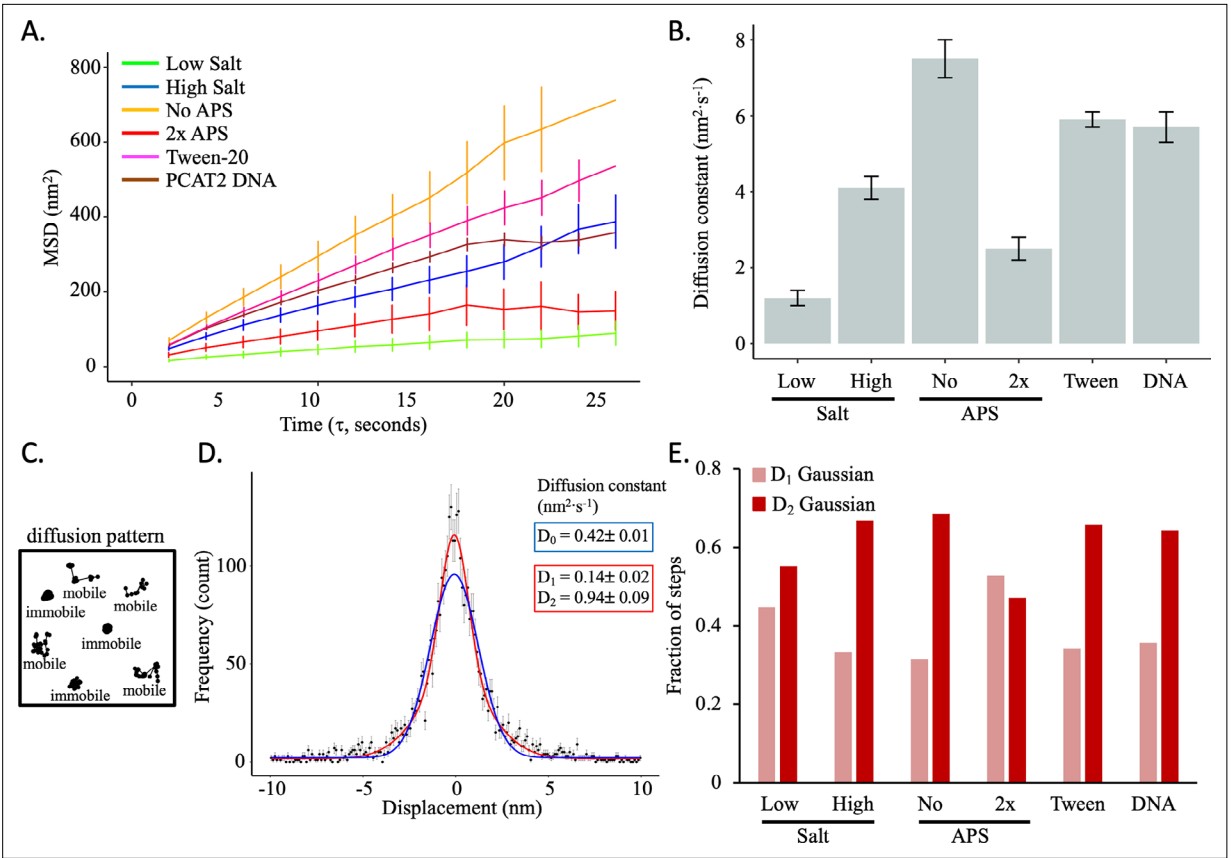

**Figure 3.** Salt and APS concentration predictably impacts CENP-A nucleosome mobility in vitro. (**A**) The average mean square displacement is shown with standard error as a function of the time interval for CENP-A nucleosome arrays in the following buffers: low salt (green; 5 mM NaCl), high salt (blue, 150 mM NaCl), no APS (yellow), twofold APS (red), 0.01% Tween-20 (pink), and PCAT2 DNA (brown). (**B**) The diffusion constants obtained from the MSD curves. The error bars represent the standard error. (**C**) Schematic representation of mobile or immobile (or paused) single particle trajectories. (**D**) The single step x-axis displacement of CENP-A nucleosomes in low salt conditions. The blue line represents a single Gaussian fit whereas the red line represents a double Gaussian fit. The latter provided a better fit to all the data for both the x- and y-step distributions for all conditions (see *Figure 3—figure supplement 5*). (**E**) The fraction of single steps corresponding to $D_1$ (narrower) Gaussian distribution or $D_2$ (wider) Gaussian distributions from the double Gaussian fitting. The $D_1$ Gaussian distribution corresponds to a smaller diffusion constant and may represent immobile or paused nucleosomes, whereas $D_2$ corresponds to a larger diffusion constant representing mobile nucleosomes. The data were obtained from two independent technical replicates per condition.

The online version of this article includes the following video, source data, and figure supplement(s) for figure 3:

**Source data 1.** Trajectory statistics for all tracked nucleosomes.

X and Y coordinates, distance between two steps, accumulated distance per trajectory, and angle between two steps are reported.

**Source data 2.** Basic statistics for all tracked nucleosomes.

**Figure supplement 1.** Impact of salt and APS concentration on step size statistics.

**Figure supplement 1—source data 1.** Trajectory statistics for all tracked nucleosomes.

X and Y coordinates, distance between two steps, accumulated distance per trajectory, and angle between two steps are reported.

**Figure supplement 1—source data 2.** Basic statistics for all tracked nucleosomes.

**Figure supplement 2.** Single (blue line) and double (red line) (sum of two) Gaussian fitting of the x and y step displacement distributions for each of the control conditions.

**Figure supplement 2—source data 1.** Trajectory statistics for all tracked nucleosomes.

X and Y coordinates, distance between two steps, accumulated distance per trajectory, and angle between two steps are reported.

**Figure supplement 2—source data 2.** Basic statistics for all tracked nucleosomes.

**Figure supplement 3.** The relative fraction of the $D_1$ and $D_2$ Gaussian fit are shown, where the $D_2$ (larger standard deviation, higher diffusion constant) Gaussian is most prevalent for all conditions except 2 x APS, Tween-20, and high salt.

**Figure supplement 4.** Individual tracts show switching between $D_1$ and $D_2$ diffusion states.

*Figure 3 continued on next page*

*Figure 3 continued*

**Figure supplement 4—source data 1.** Trajectory statistics for all tracked nucleosomes.

X and Y coordinates, distance between two steps, accumulated distance per trajectory, and angle between two steps are reported.

**Figure supplement 5.** Lower $D_1$ diffusion state and rejected "stuck" particles have different diffusion constants.

**Figure supplement 5—source data 1.** Basic statistics for the rejected 'stuck' trajectories, represented in *Figure 3—figure supplement 5*.

**Figure supplement 6.** Diffusion constant of H3 mononucleosomes higher than H3 nucleosomes within an array.

**Figure supplement 6—source data 1.** Trajectory statistics for all tracked nucleosomes.

X and Y coordinates, distance between two steps, accumulated distance per trajectory, and angle between two steps are reported.

**Figure supplement 7.** Single step distribution fit by single Gaussian for H3 mononucleosomes.

**Figure supplement 7—source data 1.** Basic statistics for all tracked nucleosomes.

**Figure supplement 8.** Schematic summary of the controls for HS-AFM quantitative single particle analysis.

**Figure 3—video 1.** HS-AFM video of in vitro reconstituted CENP-A chromatin in 5 mM NaCl containing buffer (low salt; speed =2 x).

https://elifesciences.org/articles/86709/figures#fig3video1

**Figure 3—video 2.** HS-AFM video of in vitro reconstituted CENP-A chromatin in 150 mM NaCl containing buffer (high salt; speed =2 x).

https://elifesciences.org/articles/86709/figures#fig3video2

**Figure 3—video 3.** HS-AFM video of in vitro reconstituted CENP-A chromatin imaged without functionalized mica (no APS; speed =2 x).

https://elifesciences.org/articles/86709/figures#fig3video3

**Figure 3—video 4.** HS-AFM video of in vitro reconstituted CENP-A chromatin imaged with double the amount of APS to functionalize mica (2 x APS; speed =2 x).

https://elifesciences.org/articles/86709/figures#fig3video4

**Figure 3—video 5.** HS-AFM video of in vitro reconstituted CENP-A chromatin imaged with physiological buffer +0.01% Tween (Tween; speed =2 x).

https://elifesciences.org/articles/86709/figures#fig3video5

**Figure 3—video 6.** HS-AFM video of in vitro reconstituted CENP-A chromatin on PCAT2 plasmid imaged with physiological buffer (DNA; speed =2 x).

https://elifesciences.org/articles/86709/figures#fig3video6

**Figure 3—video 7.** HS-AFM video of in vitro reconstituted H3 chromatin (speed =2 x).

https://elifesciences.org/articles/86709/figures#fig3video7

**Figure 3—video 8.** HS-AFM video of in vitro reconstituted H3 chromatin with H1.5, where H1.5 is added at a 0.2 molar ratio to H3 nucleosomes (speed =2 x).

https://elifesciences.org/articles/86709/figures#fig3video8

**Figure 3—video 9.** HS-AFM video of in vitro reconstituted H3 mononucleosomes (speed =2 x).

https://elifesciences.org/articles/86709/figures#fig3video9

might be that nucleosomes stick to the surface. To test for this possibility, we first manually verified whether individual nucleosomes could switch from what appears to be the smaller $D_1$ diffusion constant to the larger $D_2$ diffusion constant or vice versa. Indeed, we found a myriad of such examples across different conditions (*Figure 3—figure supplement 4A*). This was further reflected by the broad range of individual step sizes of each particle trajectory (*Figure 3—figure supplement 4B*, *Figure 3—source data 1*). Second, we tested whether the slower diffusing nucleosomes correspond to 'sticking' nucleosomes. To do this, we analyzed the rejected 'stuck' nucleosome trajectories with an average R-step of less than 1 nm and an R-step range of less than 8 nm. Next, we compared the effective diffusion constant of these 'stuck' nucleosomes for each video with the smaller diffusion constant obtained from the fit of a sum of two Gaussians. With the exception of the 2 x APS and low-salt conditions, we found that the effective diffusion constant of the 'stuck' nucleosome trajectories was significantly smaller than that of the smaller diffusion constant from the fit of a sum of two Gaussians (*Figure 3—figure supplement 5*, *Figure 3—figure supplement 5—source data 1*). In other words, these data would exclude the possibility of nucleosomes being 'stuck' to the mica surface.

Furthermore, the average step distribution (*Figure 3—figure supplement 1C*), maximum R-step (*Figure 3—figure supplement 1D*) and R-step range (*Figure 3—figure supplement 1E*) display a continuum of data points, instead of a bimodal distribution. Altogether, our data suggests that individual nucleosomes may have the capacity to move back and forth between the $D_1$ and the $D_2$ Gaussian distributions, indicating the possibility of switching between two diffusive modes.

Next, we expanded our analyses of the HS-AFM videos of the no APS and 2 x APS conditions (*Figure 3—videos 3 and 4*) to obtain the MSDs, and Gaussian fitting of the single step distributions. As APS functionalization positively charges the mica surface on which chromatin is deposited, we predicted that CENP-A nucleosomes trajectories would display faster dynamics in the no APS condition vs 2 x APS condition. Indeed, the slope of the MSD curve of in the absence of APS was larger compared to 2 x APS (*Figure 3A*, *Figure 3—figure supplement 1A*). The average diffusion constant was also higher in the absence of APS (7.5±0.5 nm$^2$·s$^{-1}$) than 2 x APS (2.5±0.3 nm$^2$·s$^{-1}$, *Figure 3B*, *Table 1*, *Figure 3—figure supplement 1B*, *Figure 3—source data 2*). A similar pattern was observed for the average step size (6.0±2.0 nm vs 3.8±0.2 nm, respectively), maximum R-step (15.9±0.4 nm vs 11.4±0.5 nm, respectively), and R-step variance (43±2 nm$^2$ vs 21±1 nm$^2$, respectively, *Table 1*, *Figure 3—figure supplement 1C–E*, *Figure 3—source data 2*), with higher values for no APS than 2 x APS. The single-step displacement distributions were well-fit by a sum of two Gaussians (*Figure 3—figure supplements 2 and 3*).

As an additional control, we used a very low concentration of Tween-20 (0.01%), a polysorbate surfactant that both stabilizes proteins and reduces non-specific hydrophobic interactions with the surface. We were interested to learn whether CENP-A chromatin in the presence of Tween-20 would display either more restricted nucleosomes mobility due to protein stabilization, or less restricted nucleosome mobility due to reduced non-specific interactions. HS-AFM videos of CENP-A chromatin in physiological buffer (0.5 x PBS, 2 mM MgCl$_2$) with 0.01% Tween-20 displayed the most amount of drift (*Figure 3—video 5*). After drift correction and filtering, we analyzed 587 trajectories to obtain single step distributions, MSD curves, and diffusion constants. We found that CENP-A chromatin in the presence of Tween-20 behaved more like no APS and high salt conditions with a steep MSD curve, a very broad distribution of average step sizes, and a large R-step range distribution (*Figure 3*, *Figure 3—figure supplements 1–3*, *Figure 3—source data 1 and 2*). We interpret this to mean that Tween-20 in the context of imaging CENP-A chromatin by HS-AFM primarily reduces non-specific hydrophobic interactions resulting in less restricted nucleosome mobility.

Furthermore, we wondered if our 3.5 kbp plasmid, which contains four copies of human centromere-derived α-satellite DNA, might induce nucleosome phasing or positioning (*Luger et al., 1997*; *Stormberg and Lyubchenko, 2022*). Therefore, we in vitro reconstituted CENP-A nucleosomes on an 8.5 kbp plasmid containing the lncRNA PCAT2 gene. In human cancer cell lines, ectopic CENP-A can be found at the PCAT2 locus, whereas, PCAT2 DNA is not known to position nucleosomes, in contrast to α-satellite DNA or the Widom 601-sequence (*Lowary and Widom, 1998*; *Luger et al., 1997*; *Stormberg and Lyubchenko, 2022*; *Thåström et al., 1999*). We observed an MSD curve similar to the high salt conditions (*Figure 3A*, *Figure 3—video 6*, *Figure 3—source data 1 and 2*), with an average diffusion constant of 5.7±0.4 nm$^2$·s$^{-1}$ (*Figure 3B*, *Table 1*, *Figure 3—source data 1 and 2*). A similar pattern was observed for the average step size (5.8±0.2 nm), maximum R-step (16.0±0.2 nm), and R-step variance (33.7±0.6 nm$^2$, *Table 1*, *Figure 3—figure supplement 1C-E*, *Figure 3—source data 2*). The single step displacement distributions were well-fit by a sum of two Gaussians (*Figure 3—figure supplement 2*, *Figure 3—source data 1 and 2*). Overall, it appears that CENP-A nucleosomes reconstituted on the 8.5 kbp plasmid without known positioning sequences behaves similar to CENP-A nucleosomes reconstituted on 3.5 kbp plasmid with known positioning sequences, thereby suggesting that nucleosome mobility measured in these experiments may be independent of DNA sequence specificity.

When we analyzed previously published HS-AFM videos of H3 chromatin with or without linker histone H1.5 (*Melters and Dalal, 2021*) as well as H3 mononucleosomes, we observed a bias in both the angle between successive steps and Gaussian fitting of single step displacements (*Figure 3—figure supplements 6 and 7*, *Figure 3—videos 7–9*, *Figure 3—source data 1 and 2*). These data provide evidence that bias in HS-AFM trajectories is a possibility and that it can be detected. Furthermore, there was no difference in fitting either a single or double Gaussian distributions for H3 mononucleosomes (*Figure 3—figure supplement 7F*). Mononucleosomes are not associated with other nucleosomes and a priori mononucleosomes cannot display whole chromatin fiber motions, allowing mononucleosomes to move freely independent of the DNA strand. We observed that the diffusion constant of mononucleosomes is about threefold larger than that of chromatin arrays (*Table 1*) and 10- to 185-fold larger than the D$_1$ diffusion constants observed under various conditions (*Figure 3—figure supplements 2 and 7*). In addition, the R-step range of mononucleosomes is roughly twofold

larger than chromatin arrays (*Table 1*). Therefore, the latter observations imply that the unconstrained motion of mononucleosomes results in a single Gaussian distribution of step displacements.

Altogether, when we analyzed HS-AFM videos, CENP-A chromatin responded to varying salt and APS concentrations in agreement with previous reports (*Allahverdi et al., 2015*; *Brasch et al., 1971*; *Lyubchenko et al., 2014*; *Shlyakhtenko et al., 2003*; *Yager et al., 1989*; *Yager and van Holde, 1984*). Low salt and 2 x APS concentrations reduced CENP-A nucleosome mobility, whereas high salt and no APS concentrations increased CENP-A nucleosome mobility (*Figure 3—figure supplement 8*). For the remainder of the HS-AFM experiments, we used near physiological relevant salt concentrations (67.5 mM NaCl, 2 mM $MgCl_2$) and standardized APS concentrations (167 nM [*Lyubchenko et al., 2014*]).

## CENP-C$^{CD}$ represses CENP-A nucleosome mobility in vitro

Previously, we showed that a central domain fragment of CENP-C (*Figure 4A*) rigidified CENP-A nucleosomes and induced CENP-A nucleosome clustering (*Melters et al., 2019*). Based on these observations, we hypothesized that CENP-C$^{CD}$ would reduce CENP-A nucleosome mobility. To test this hypothesis, we imaged CENP-A chromatin by HS-AFM under near physiological conditions. Subsequently, nucleosome tracks were extracted in either the absence (*Figure 4B*, *Figure 4—figure supplements 1–3*, *Figure 4—video 1*, *Figure 4—source data 1 and 2*) or presence of 1, 2, or 4 CENP-C$^{CD}$ molecules (1 x, 2 x, or 4 x, respectively) per CENP-A nucleosome (*Figure 4C*, *Figure 4—figure supplements 1–3*, *Figure 4—videos 2–4*, *Figure 4—source data 1 and 2*). From at least 3 experiments per sample, we obtained 498, 368, 310, and 166 trajectories, respectively (*Table 1*). First, we verified that there were no motion artifacts associated with the tip scanning (*Figure 4—figure supplement 1*). Next, we calculated the MSD of CENP-A nucleosomes alone or in the presence of 1 x, 2 x, or 4 x CENP-C$^{CD}$ and found that individual MSD curves of CENP-A nucleosomes were broadly distributed (*Figure 4D*, *Figure 4—figure supplement 2A*, B), with an average MSD curve that reached a plateau after ~25 s (*Figure 4—figure supplement 2A*), implying confined motion (*Kapanidis et al., 2018*; *Zhong and Wang, 2020*).

When CENP-C$^{CD}$ was added at either 2 x or 4 x molar excess to CENP-A nucleosomes, CENP-A nucleosome mobility was strongly restricted (*Figure 4D*, *Figure 4—figure supplement 2A*, *Figure 4—videos 2–4*, *Figure 4—source data 1 and 2*). Indeed, the MSD curve was much lower for 2 x and 4 x CENP-C$^{CD}$ compared to CENP-A alone, and the 2 x and 4 x CENP-C$^{CD}$ MSD curve maintained a shallow slope (*Figure 4D*, *Figure 4—figure supplement 2A*). In contrast, the 1 x CENP-C$^{CD}$ MSD curve was similar to CENP-A alone (*Figure 4D*, *Figure 4—figure supplement 2B*, *Figure 4—source data 1 and 2*). The diffusion constant of CENP-A alone was 2.3±0.2 nm$^2 \cdot$s$^{-1}$, whereas the addition of 2 x and 4 x CENP-C$^{CD}$ reduced the diffusion constant 2.9-fold to 0.78±0.06 nm$^2 \cdot$s$^{-1}$ and 3.8-fold to 0.61±0.05 nm$^2 \cdot$s$^{-1}$, respectively (*Table 1*, *Figure 4E*, *Figure 4—figure supplement 2B*, *Figure 4—source data 1 and 2*). This difference is reflected in the smaller single frame step size when CENP-C$^{CD}$ is added to CENP-A chromatin compared to CENP-A chromatin alone (*Figure 4—figure supplement 2C*, *Figure 4—source data 2*). The maximum R-step and R-step range were also larger for CENP-A nucleosomes without CENP-C$^{CD}$ (*Figure 4—figure supplement 2D*, E, *Figure 4—source data 2*). Next, we fitted the single step displacement distributions with Gaussian distributions and found that a sum of two Gaussian distributions provided better fits (*Figure 4—figure supplement 3*, *Figure 4—source data 1 and 2*). Overall, by HS-AFM we observed that CENP-C$^{CD}$ restricts CENP-A nucleosome mobility in a switch-like manner. We hypothesized that this switch, as we recently observed (*Melters et al., 2019*), could be the mechanism by which overexpression of CENP-C in living cells results in CENP-A chromatin clustering.

## Excess CENP-C suppresses centromeric RNAP2 levels and centromeric transcription in vivo

Next, we asked what the functional consequences are of CENP-C on CENP-A chromatin, beyond the formation of kinetochores (*Cheeseman et al., 2006*; *Walstein et al., 2021*; *Mendiburo et al., 2011*; *Régnier et al., 2005*). Previously, we showed that overexpressing CENP-C in HeLa cells resulted in increased clustering of centromeric chromatin and loss of centromeric RNAP2 (*Melters et al., 2019*). These results, combined with our 'decrease of motion' observations above, suggest that centromeric non-coding α-satellite transcription might be impaired in the background of CENP-C overexpression.

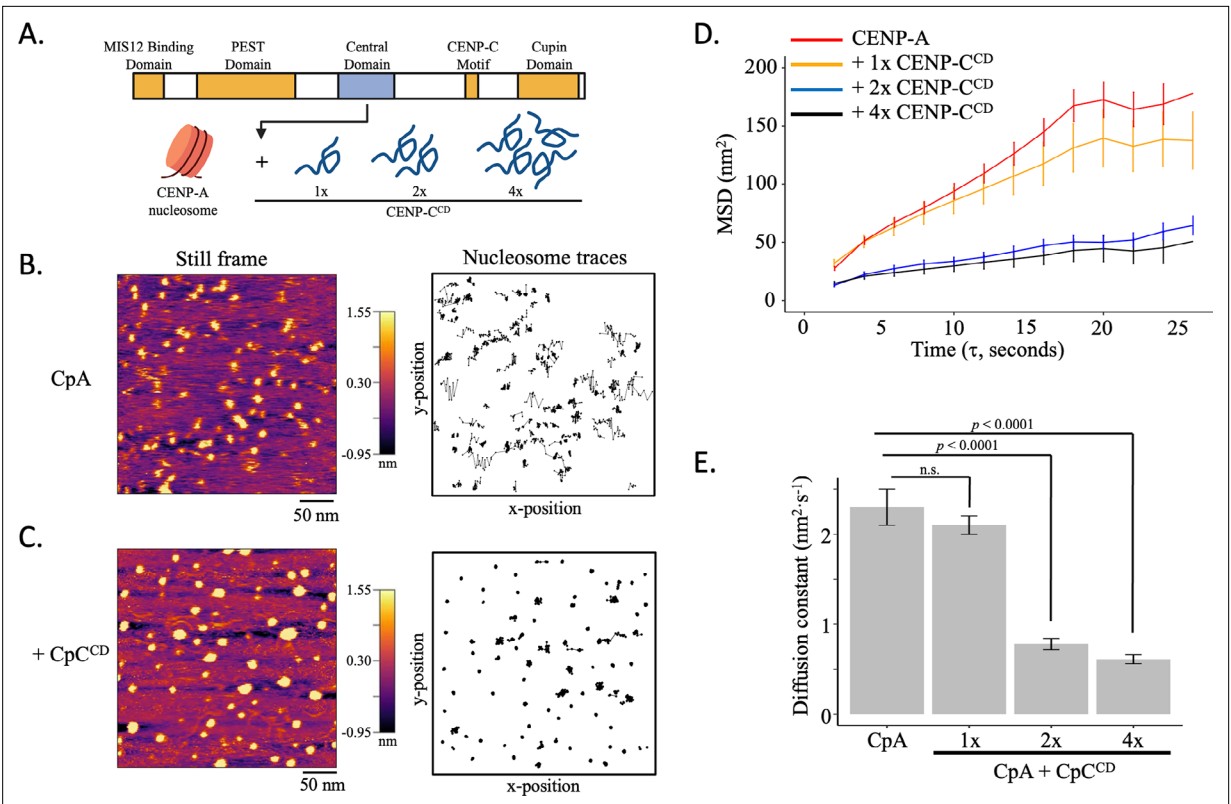

**Figure 4.** CENP-C$^{CD}$ restrict CENP-A nucleosome mobility in vitro (**A**) Schematic representation of human CENP-C (943 amino acids: CENP-C, NCBI Gene ID: 1060). The central domain of CENP-C (CENP-C$^{CD}$), which directly binds to CENP-A nucleosomes, was added at a ratio of 1, 2, or 4 fragments per CENP-A nucleosome (1 x, 2 x, or 4 x CENP-C$^{CD}$, respectively). (**B**) CENP-A nucleosome arrays were tracked in fluid for up to 120 s by HS-AFM at 1 frame every 2 s. A representative still frame is shown as well as the trajectories over time. (**C**) CENP-A nucleosome arrays were tracked in the presence of 1 x, 2 x, or 4 x CENP-C$^{CD}$. A representative still frame is shown as well as the trajectories over time. (**D**) The average mean square displacement is shown with standard error as a function of the time interval. CENP-A nucleosomes alone are in red. CENP-A nucleosomes with 1 x CENP-C$^{CD}$ is in yellow, 2 x CENP-C$^{CD}$ is in blue, and 4 x CENP-C$^{CD}$ is in black. (**E**) The diffusion constants obtained from the MSD curves. The line and bar graphs represent three independent technical replicates. The error bars represent the standard error.

The online version of this article includes the following video, source data, and figure supplement(s) for figure 4:

**Source data 1.** Trajectory statistics for all tracked nucleosomes.

X and Y coordinates, distance between two steps, accumulated distance per trajectory, and angle between two steps are reported.

**Source data 2.** Basic statistics for all tracked nucleosomes.

**Figure supplement 1.** The distribution of the angle between successive nucleosome positions for CENP-A chromatin alone or in the presence of 1 x, 2 x, or 4 x CENP-C$^{CD}$.

**Figure supplement 2.** CENP-C$^{CD}$ restricts step size of CENP-A nucleosomes.

**Figure supplement 2—source data 1.** Trajectory statistics for all tracked nucleosomes.

**Figure supplement 2—source data 2.** Basic statistics for all tracked nucleosomes.

This includes diffusion constant (nm$^2$ ·s$^{-1}$), MSD slope (nm$^2$ ), average MSD slope (nm$^2$), average step size (nm), maximum R-step (nm), and R- step range (nm).

**Figure supplement 3.** CENP-C$^{CD}$ does not impact double Gaussian of single step distribution of CENP-A nucleosomes.

**Figure supplement 3—source data 1.** Trajectory statistics for all tracked nucleosomes.

X and Y coordinates, distance between two steps, accumulated distance per trajectory, and angle between two steps are reported.

**Figure supplement 3—source data 2.** Basic statistics for all tracked nucleosomes.

**Figure 4—video 1.** HS-AFM video of in vitro reconstituted CENP-A chromatin (speed =2 x).

https://elifesciences.org/articles/86709/figures#fig4video1

**Figure 4—video 2.** HS-AFM video of in vitro reconstituted CENP-A chromatin with CENP-C$^{CD}$ where CENP-C$^{CD}$ is added at a 1 x molar ratio to CENP-A nucleosomes (speed =2 x).

*Figure 4 continued on next page*

*Figure 4 continued*

https://elifesciences.org/articles/86709/figures#fig4video2

**Figure 4—video 3.** HS-AFM video of in vitro reconstituted CENP-A chromatin with CENP-C$^{CD}$ where CENP-C$^{CD}$ is added at a 2 x molar ratio to CENP-A nucleosomes (speed =2 x).

https://elifesciences.org/articles/86709/figures#fig4video3

**Figure 4—video 4.** HS-AFM video of in vitro reconstituted CENP-A chromatin with CENP-C$^{CD}$ where CENP-C$^{CD}$ is added at a 4 x molar ratio to CENP-A nucleosomes (speed =2 x).

https://elifesciences.org/articles/86709/figures#fig4video4

To examine this facet of CENP-C:CENP-A homeostasis, we overexpressed CENP-C in HeLa cells to assess if centromeric transcription is altered.

First, we measured the effects of CENP-C overexpression on the level of RNAP2 on CENP-A chromatin. We overexpressed CENP-C 2.8-fold (*Figure 5—figure supplement 1*; *Figure 5—figure supplement 1—source data 1 and 2*) in HeLa cells for 72 hr and subsequently purified CENP-A chromatin associated with CENP-C by CENP-C native ChIP (nChIP), and the unbound CENP-A chromatin was pulled-down by sequential ACA nChIP. We found that RNAP2 levels were reduced upon CENP-C overexpression (*Figure 5A*, *Figure 5—figure supplement 1*, *Figure 5—figure supplement 1—source data 1 and 2*). In addition, we observed that, upon CENP-C overexpression, total CENP-A levels were also reduced (*Figure 5B*, *Figure 5—figure supplement 1*, *Figure 5—figure supplement 1—source data 1 and 2*). Previously (*Melters et al., 2019*), we showed that the addition of CENP-C$^{CD}$ or overexpression of CENP-C results in compaction of CENP-A chromatin. We therefore wondered if CENP-C overexpression impacted centromeric transcription. By quantitative PCR, we observed

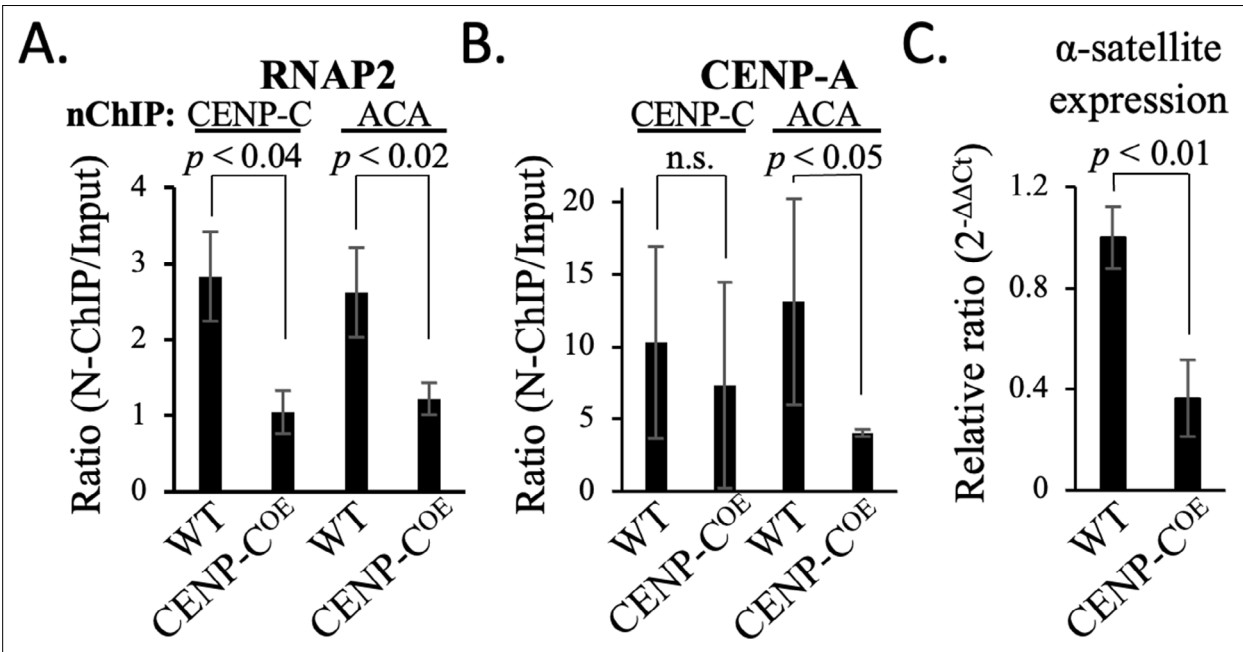

**Figure 5.** CENP-C overexpression suppressed α-satellite expression and centromeric RNAP2 occupancy. (**A**) Quantification of RNAP2 levels pulled down with either CENP-C or sequential ACA nChIP. (**B**) Quantification of CENP-A levels that pulled down with either CENP-C or sequential ACA nChIP. (**C**) Quantification of consensus α-satellite transcription in mock-transfected (WT) and CENP-C overexpression (CENP-C$^{OE}$) (two-sided t-test; significance was determined at p<0.05). The bar graphs represent three independent technical replicates, and the error bars represent standard deviations.

The online version of this article includes the following source data and figure supplement(s) for figure 5:

**Source data 1.** Quantification of RT-PCR of α-satellite transcripts for *Figure 5C*.

**Figure supplement 1.** Representative western blot quantifying RNAP2 and CENP-A levels.

**Figure supplement 1—source data 1.** Entire western blot as shown in Figure 5 – figure supplement 1.

**Figure supplement 1—source data 2.** Quantification of ChIP-western blot for Figures 5A, B.

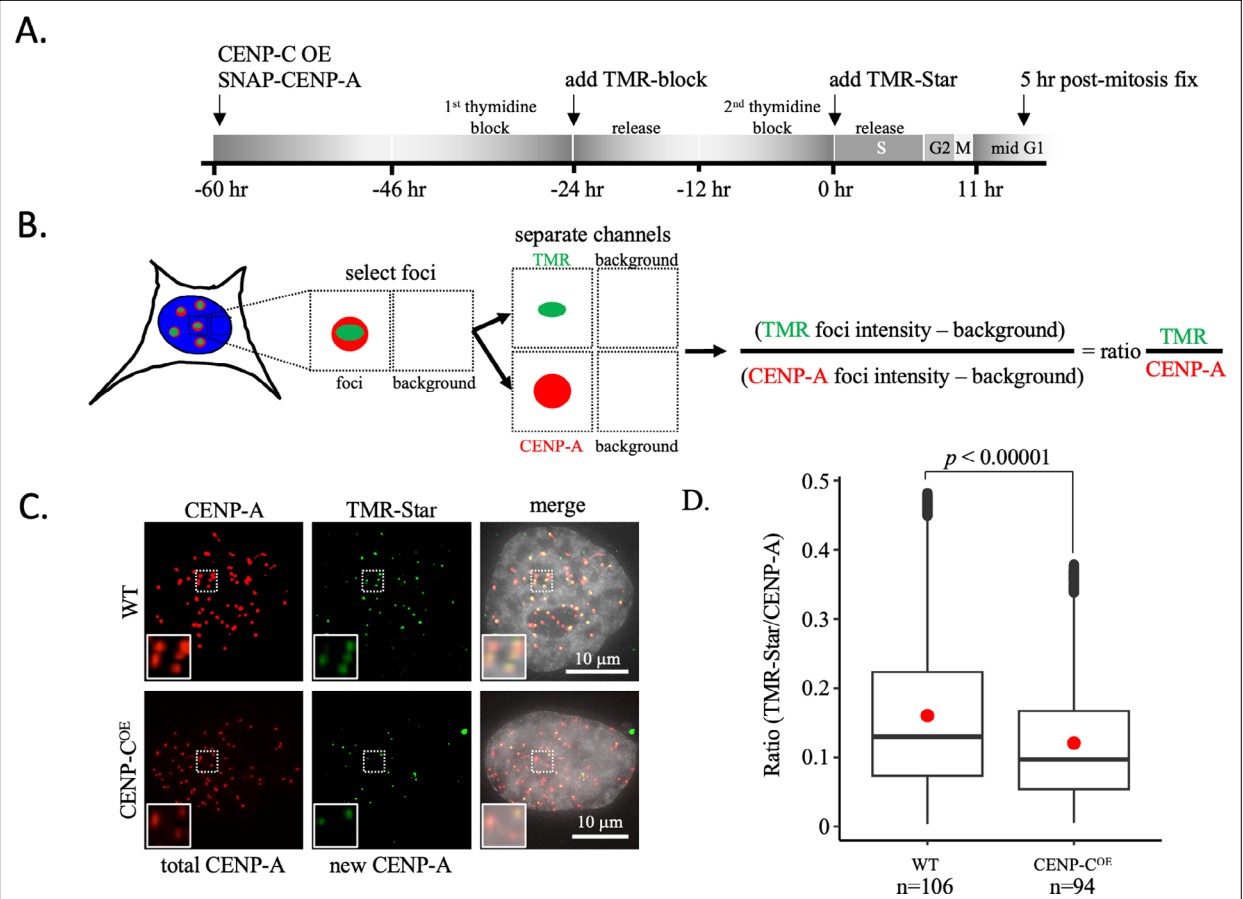

**Figure 6.** New CENP-A loading impaired upon CENP-C overexpression. (**A**) Schematic of experimental design. (**B**) Colocalized immunofluorescent signals for CENP-A and TMR-Star are collected and the intensity of both foci is measured as well as the background neighboring the foci to determine the ratio of the TMR-star signal over total CENP-A signal. (**C**) De novo CENP-A incorporation was assessed by quench pulse-chase immunofluorescence. After old CENP-A was quenched with TMR-block, newly loaded CENP-A was stained with TMR-Star and foci intensity was measured over total CENP-A foci intensity. Inset is a ×2 magnification of the dotted box in each respective image. (**D**) Quantification of de novo CENP-A loading by measuring the ratio of TMR-Star signal over total CENP-A signal (one-way ANOVA; significance was determined at p<0.05, n=number of cells analyzed). The box plots represent three independent technical replicates.

The online version of this article includes the following source data and figure supplement(s) for figure 6:

**Figure supplement 1.** Quantification of de novo CENP-A loading by measuring the ratio of TMR-Star signal over total CENP-A signal as shown in *Figure 6C* with individual data points.

**Figure supplement 1—source data 1.** Quantification of de novo CENP-A loading by measuring background corrected foci intensity for WT and CENP-C overexpressed cells.

a ~60% reduction in α-satellite transcripts in cells overexpressing CENP-C (*Figure 5C*, *Figure 5— source data 1*).

These results indicate that CENP-A levels are reduced and that centromeric transcription is indeed impaired upon CENP-C overexpression. Previous reports showed that new CENP-A loading is transcriptionally regulated (*Jansen et al., 2007*; *Quénet and Dalal, 2014*). Therefore, we hypothesized that CENP-C overexpression leads to defective de novo CENP-A loading.

## CENP-C overexpression limits de novo CENP-A loading

To test this hypothesis, we turned to the well-established SNAP-tagged CENP-A system combined with quench pulse-chase immunofluorescence (*Bodor et al., 2012*). Using this system in cells synchronized to mid-G1, one can distinguish between older CENP-A (TMR-block) and newly incorporated CENP-A (TMR-Star; *Figure 6A and B*). Strikingly, in an CENP-C overexpression background, we

observed a 1.3-fold reduction of de novo incorporation of CENP-A (*Figure 6C and D*, *Figure 6—figure supplement 1—source data 1*).

Thus, the functional consequences of CENP-C overexpression are suppression of α-satellite transcription (*Figure 5C*) and subsequent impairment of new CENP-A loading (*Figure 6C and D*).

## Discussion

Here, we demonstrate that HS-AFM is a reliable technique for studying single nucleosome dynamics by directly visualizing their motion in real time. Importantly, we show that the AFM tip scanning motion does not generate observable artifacts in nucleosome dynamics (*Figure 2*, *Figure 2—figure supplements 1 and 2*). Various critical scanning conditions were tested to assess their impact on CENP-A chromatin dynamics. Salt is a well-known factor that can either stabilize or destabilize chromatin (*Allahverdi et al., 2015*; *Brasch et al., 1971*; *Lyubchenko et al., 2014*; *Shlyakhtenko et al., 2003*; *Yager et al., 1989*; *Yager and van Holde, 1984*). Case in point, salt dialysis is a common method for in vitro nucleosome reconstitution (*Cruz-Becerra and Kadonaga, 2021*; *Peterson, 2008*; *Walkiewicz et al., 2014*). Indeed, low salt restricted CENP-A nucleosome motion, whereas high salt did the opposite (*Figure 3*, *Figure 3—figure supplements 1–3*). In AFM studies, APS is used to functionalize the mica surface with positive charges facilitating the interaction of DNA or chromatin with the mica surface. Not functionalizing the mica surface resulted in more mobile CENP-A nucleosomes, whereas 2 x APS functionalized mica resulted in increased levels of immobile CENP-A nucleosomes (*Figure 3*, *Figure 3—figure supplements 1–3*). Furthermore, we used a plasmid containing a non-nucleosome positioning sequence PCAT2 (*Figure 3*, *Figure 3—figure supplements 1–3*). Interestingly, the diffusion constant for CENP-A nucleosomes reconstituted on PCAT2 containing plasmid was higher than that of CENP-A nucleosomes reconstituted on α-satellite containing plasmid imaged under the same conditions (*Table 1*).There are various parameters that could account for this difference in diffusion constant. Only 20% of the α-satellite containing plasmid is comprised of nucleosome positioning sequences. Although it cannot be excluded, it is unlikely that the nucleosome positioning sequence reduces the global diffusion constants. Alternatively, the two plasmids differed substantially in length (8.5 kbp PCAT2 plasmid versus 3.5 kbp α-satellite plasmid). This means that PCAT2 plasmid possibly contained a much larger number of nucleosomes, potentially impacting local nucleosome density and thus chromatin compaction. The potential role of nucleosome crowding on nucleosome mobility should be investigated. In addition, H3 mononucleosomes were the only nucleosomes for which the single-frame step size distribution was well-fit by a single Gaussian (*Figure 3—figure supplement 7F*), this is in stark contrast with CENP-A chromatin, for which the step size distribution is well-fit by the sum of two Gaussians (*Figure 3—figure supplement 2*, *Figure 4—figure supplement 2*). This implies that nucleosome arrays might drive the two mobility states we observed. Looking more closely at individual tracks, we noticed that nucleosomes can switch between the $D_1$ and $D_2$ state (*Figure 3—figure supplement 3*). The slower $D_1$ diffusion constant did not appear to simply reflect nucleosomes being stuck to the mica surface (*Figure 3—figure supplement 5*). It will be interesting to learn what precisely causes the two Gaussian distributions, including whether oncohistones (*Nacev et al., 2019*), PTMs, or nucleosome binding factors (*Zhou et al., 2019*) can alter the relative probability of the two Gaussian distributions. Altogether, these data provide evidence that CENP-A nucleosomes respond to various control conditions in a predictable manner, but also that their dynamics are complex and include at least two mobility states.

CENP-C modulates both CENP-A nucleosome accessibility (*Ali-Ahmad et al., 2019*; *Ariyoshi et al., 2021*; *Falk et al., 2016*; *Falk et al., 2015*; *Guo et al., 2017*) and its elasticity (*Melters et al., 2019*). Here, we show that CENP-C$^{CD}$ regulates CENP-A nucleosomes mobility in a switch-like manner (*Figure 4*, *Figure 4—figure supplements 1–3*). This might imply that CENP-C compacts CENP-A chromatin once a critical mass is reached. The data presented here correlates with CENP-C$^{CD}$ rigidification of CENP-A nucleosomes (*Melters et al., 2019*). Additionally, we show that in vivo, CENP-C overexpression results in reduced levels of centromeric RNAP2 (*Melters et al., 2019*), impaired centromeric transcription (*Figure 5C*), and subsequent decreased loading of new CENP-A at the centromere (*Figure 6*). These findings combined provide evidence for a speculative link between the physical properties of nucleosomes, their mobility along DNA, and how these material properties might regulate chromatin accessibility and transcriptional potential.

In the nucleus, nucleosomes move in different dimensions, either in a single dimension along the DNA strand, or three dimensions where the DNA strand moves and the nucleosomes follow as passengers (*Babokhov et al., 2020*; *Ide et al., 2022*; *Melters and Dalal, 2021*). In addition, various events, ranging from transcription to DNA repair and replication, involve chromatin remodeling (*Nodelman and Bowman, 2021*; *Zaret, 2020*). Several in vivo studies have utilized tagged H2B combined with high-resolution live cell imaging to probe nucleosome dynamics. Nucleosomes within heterochromatin were less dynamic compared to euchromatin regions (*Ashwin et al., 2019*; *Maeshima et al., 2023*; *Nozaki et al., 2017*; *Ricci et al., 2015*), transcriptional inhibition diminished constraints of local chromatin movements (*Nagashima et al., 2019*), and local levels of H1 correlated with reduced nucleosome dynamics (*Gómez-García et al., 2021*). Indeed, chromatin-associated H1-eGFP mobility was two orders of magnitude smaller than free H1-eGFP (*Bernas et al., 2014*; *Wachsmuth et al., 2016*). ATP-driven effects also impact DNA motion (*Bhattacharya et al., 2006*; *Levi et al., 2005*). Tagging DNA is an effective method to test how mobile chromosome loci are in vivo. Using fluorescence recovery spectroscopy, GFP-tagged loci in CHO cells showed a slow and fast diffusion coefficient of $0.24 \cdot 10^{-3}$ and $3.13 \cdot 10^{-3}$ $\mu m^2 \cdot s^{-1}$ (*Levi et al., 2005*) and TetO tagged-DNA $D_H J_H$ loci in B cells had a diffusion coefficient of ~$2.0 \cdot 10^3$ $\mu m^2 \cdot s^{0.5}$ (*Lucas et al., 2014*). The change in diffusion coefficient of $V_H D_H$ loci is interpreted to results in a fourfold increase in interaction frequency (*Lucas et al., 2014*). In these cases, specific loci were tagged and tracked, but the behavior of individual nucleosomes within these domains remain unaccounted for. Therefore, in parallel, studies using optical tweezers probed the one dimensional diffusive behavior of nucleosomes (*Chen et al., 2019*; *Rudnizky et al., 2019*). A recent study using optical tweezers showed that the diffusion constant of H3 mononucleosomes sliding along a DNA strand is ~1.3 $bp^2 \cdot s^{-1}$ (~0.15 $nm^2 \cdot s^{-1}$; *Rudnizky et al., 2019*). This value is roughly similar to the $D_1$ diffusion constants we observed (*Figure 3—figure supplements 2 and 7*, *Figure 4—figure supplements 1 and 2*), implying that the $D_1$ diffusion constant might reflect sliding nucleosomes. Binding of transcription factors to nucleosomal DNA biases nucleosome motion, potentially through manipulation of the folding and unfolding of nucleosomal DNA around the nucleosome core particle (*Donovan et al., 2023*; *Rudnizky et al., 2019*). Here, we show that CENP-A nucleosomes have an average diffusion constant of 2.3±0.2 $nm^2 \cdot s^{-1}$ and their single-step distribution was well-fit by a sum of two Gaussian distributions (*Figure 4*, *Figure 4—figure supplement 2*). When CENP-C$^{CD}$ was added at a two- or fourfold molar ratio to CENP-A nucleosomes, we observed a diffusion constant that is about three times lower than CENP-A nucleosomes alone (0.78±0.06 $nm^2 \cdot s^{-1}$ and 0.61±0.05 $nm^2 \cdot s^{-1}$, respectively). This difference in diffusion constants for nucleosomes between the Rudnizky study (~0.15 $nm^2 \cdot s^{-1}$, [*Rudnizky et al., 2019*]) and our results (0.61–2.3 $nm^2 \cdot s^{-1}$) could be due to technical differences of their experimental set-up compared to our experimental approach (optical tweezers vs HS-AFM). We used nucleosome arrays which permit both one-dimensional (nucleosome sliding) and two-dimensional motions (whole chromatin fiber movement) of nucleosomes compared to mononucleosomes that were fixed to polystyrene beads, which only permit one-dimensional motion (*Rudnizky et al., 2019*). In contrast, single molecule tracking in cultured cells captures nucleosome dynamics within the nucleus (*Iida et al., 2022*; *Kimura and Cook, 2001*; *Morisaki et al., 2014*; *Wagh et al., 2023*), but requires fluorophore-tags. Fluorophore-tags are photosensitive (*Jradi and Lavis, 2019*) and have the potential of altering the function of the protein it is bound to *Maheshwari et al., 2015*; *Ravi et al., 2010*. Calculated diffusion constants for H2B-eGFP range from 0.0019 $\mu m^2 \cdot s^{-1}$ to 7.3 $\mu m^2 \cdot s^{-1}$ (*Bhattacharya et al., 2006*; *Mazza et al., 2012*) and 0.03 $\mu m^2 \cdot s^{-1}$ to 1.39 $\mu m^2 \cdot s^{-1}$ for HALO-H2B (*Lovely et al., 2020*; *Ranjan et al., 2020*). These in vivo H2B diffusion constants are several orders of magnitudes larger than we observed (0.61–2.3 $nm^2 \cdot s^{-1}$). This could be due to the many activities within the nucleus of a living cell compared to the steady-state nature of an in vitro experiment. As such, the potential of single molecule analysis by HS-AFM to contribute to elegant studies in the field is exciting, as it can link single-molecule force spectroscopy analysis to in vivo single molecule tracking experiments.

Looking at another histone variant nucleosome, the diffusion constant of H2A.Z nucleosomes is larger than H3 nucleosomes (*Rudnizky et al., 2019*), which correlates with the transcriptional buffering function of H2A.Z (*Chen et al., 2019*; *Giaimo et al., 2019*). A logical prediction from our results would be that transcription through CENP-A chromatin would be more efficient compared to H3 chromatin. Indeed, a recent single-molecule study showed that CENP-A creates an open chromatin structure (*Nagpal and Fierz, 2022*). We found that proteins that exert their activity by binding to the outer

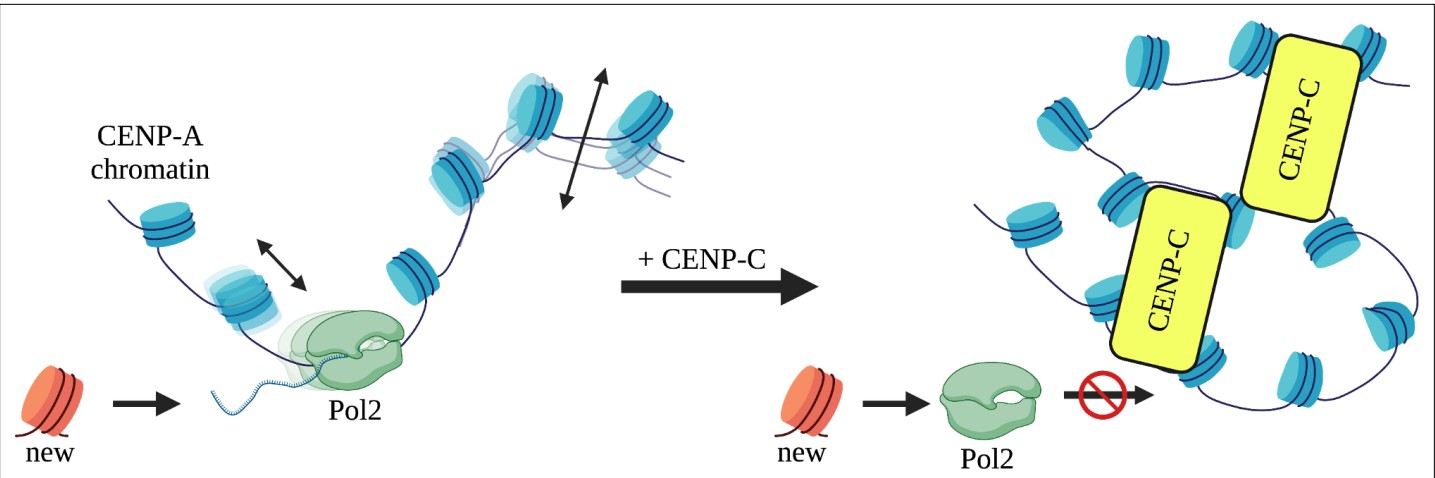

**Figure 7.** Clutch model for CENP-C restricted motion of CENP-A chromatin. Under wildtype conditions, we propose that CENP-A chromatin not bound by CENP-C (yellow box) forms a chromatin clutch and is readily accessible to the transcriptional machinery, because of the intrinsic material properties of CENP-A nucleosomes. In contrast, when CENP-C or CENP-C complexes bind CENP-A nucleosomes, a unique clutch of CENP-A chromatin is formed restricting sliding of CENP-A nucleosomes. This coincides with CENP-C^CD altering the material properties, and quenching the mobility, of CENP-A chromatin. Less mobile CENP-A nucleosomes restrict progression of the transcriptional machinery and subsequent loading of new CENP-A molecules.

surface of nucleosomes, rather than changing the internal constituents of nucleosomes, have the opposite effect on transcription. CENP-C^CD rigidifies CENP-A nucleosomes (*Melters et al., 2019*), and CENP-C^CD-CENP-A nucleosomes have a significantly reduced diffusion constant compared to CENP-A nucleosomes alone (*Table 1*). In vivo, overexpression of CENP-C resulted in decreased centromeric transcription (*Figure 6*). Thus, this study significantly extends the existing paradigm that suggests nucleosome dynamics are highly tunable, with a surprising twist – tuners can dampen or exaggerate nucleosome motion, which correlates with higher order chromatin folding and accessibility.

Taking these and prior findings in context, our data suggest a model where chromatin effector partners modify the material properties of histone variant nucleosomes at the local level, supporting the formation of functional nucleosome clutches (*Portillo-Ledesma et al., 2021*; *Ricci et al., 2015*; *Figure 7*). Here, we specifically probed CENP-A chromatin and based on our results we propose the following working model. CENP-A nucleosomes recruit CENP-C to form a kinetochore-promoting CENP-A chromatin clutch (*Kale et al., 2023*). CENP-C functions as the template for the recruitment of additional kinetochore proteins (*Klare et al., 2015*; *Yatskevich et al., 2023*). When CENP-C binds, CENP-A nucleosomes become rigidified (*Melters et al., 2019*) and restrict CENP-A nucleosomes mobility (*Figure 4*). This unique clutch facilitates the recruitment of other inner kinetochore components by locally immobilizing CENP-A chromatin (*Yatskevich et al., 2023*). Yet, to load new CENP-A molecules, transcription must happen (*Arunkumar and Melters, 2020*; *Quénet and Dalal, 2014*). The repressive chromatin state that CENP-C induces (*Figure 5*) contradicts this functional necessity. One speculative manner in which cells can work around the juxtaposition of the dual functions of CENP-A chromatin is by maintaining a pool of free centromeric CENP-A nucleosomes (*Figure 7*). Indeed, ChIP-seq and FISH data have established that centromeric CENP-C levels are lower compared to CENP-A levels (*Henikoff et al., 2015*; *Kyriacou and Heun, 2018*). We propose that unbound elastic and mobile CENP-A chromatin clutches create an intrinsically accessible chromatin state, allowing for the recruitment of transcriptional machinery that maintains centromeric CENP-A levels to facilitate both opposing functions (*Figure 7*). Succinctly, the epigenetic fate of a locus may be tightly, possibly causally, linked to the mechanical state of the chromatin fiber; concomitantly, the reinstatement of an epigenetic signature by de novo loading of a particular variant, reinforces the mechanical state of the fiber.

In summary, we report the MSDs and diffusion constants for CENP-A nucleosomes under a variety of conditions, and we show that single-step distributions are well-fit by the sum of two Gaussians. This implies that nucleosomes exist in at least two distinct mobility states. CENP-C^CD modulates centromeric chromatin by restricting the motions of CENP-A nucleosomes. As we observed a switch-like behavior of restricting CENP-A nucleosome motions, instead of a linear dose-response, we speculate

that CENP-C limits CENP-A mobility once a critical concentration is reached. In vivo, we observed diminished transcriptional competence when CENP-C is overexpressed, resulting in suppression of de novo CENP-A loading. These results strongly imply that there is a mechanistic link between modulating the material properties of nucleosomes and chromatin accessibility. Nucleosome dynamics play an important role in genome compaction and regulating DNA access by DNA binding factors. These dynamics are driven by only a few interactions between the interfaces of DNA and nucleosomes (*Fierz and Poirier, 2019*; *Polach and Widom, 1995*; *Widom, 1998*). An exciting line of investigation is to examine how at the nanoscale level the interaction between CENP-A nucleosomes and CENP-C protein co-evolved to repress CENP-A's mobile and elastic nature. It will also be crucial to examine how CENP-A:CENP-C homeostasis, along with those of other key inner kinetochore proteins such as CENP-N, T, S, X, and W regulate centromeric transcription and thereby de novo assembly of CENP-A to maintain the epigenetic identity of centromeres in other species. At a more global scale, it will be exciting to ask how different H1 variants modulate the chromatin fiber at the local level to promote or limit transcriptional competency, and how H1 variants dictate the mechanical motions of individual nucleosomes within the local 10 nm chromatin fiber.

# Materials and methods

## Key resources table

| Reagent type (species) or resource | Designation | Source or reference | Identifiers | Additional information |
|---|---|---|---|---|
| Cell line (*Homo sapiens*) | HeLa (cervical carcinoma, Adult) | ATCC | CCL-2 | |
| Antibody | Anti-CENP-A (Mouse monoclonal) | Abcam | Cat. #: ab13939, RRID: AB_300766 | IF(1:1000) |
| Antibody | Anti-CENP-A (rabbit monoclonal) | Abcam | Cat. #: ab45694 | WB (1:3000) |
| Antibody | Anti-CENP-C (guinea pig polyclonal) | MBL International | Cat. #: PD030, RRID: AB_10693556 | nChIP (5 μL), WB (1:1000) |
| Antibody | Anti-CENP-C (rabbit polyclonal) | Santa Cruz | Cat. #: sc-22789 | WB (1:500) |
| Antibody | Anti-H2A (rabbit polyclonal) | Abcam | Cat. #: ab18255, RRID: AB_470265 | WB (1:1000) |
| Antibody | Anti-CENP-A (rabit polyclonal) | This paper | | nChIP (3 μL per test) |
| Antibody | ACA serum (human, polyclonal) | BBI Solutions | SG140-2 | nChIP (5 μL per test) |
| Recombinant DNA reagent | GFP-CENP-C (plasmid) | Gift from Stephan Diekmann | | pGFP-CENP-C (KAN) |
| Recombinant DNA reagent | SNAP-CENP-A (plasmid) | This paper | | SNAP version of pCh-C-CENP-A (AMP) |
| Recombinant DNA reagent | PCAT2 (plasmid) | *Arunkumar et al., 2022* | | |
| Recombinant DNA reagent | 4 x α-satellite (plasmid) | *Quénet and Dalal, 2014* | | |
| Sequence-based reagent | Centromeric α-satellite_F | *Quénet and Dalal, 2014* | PCR primers | CATCACAAAGAAGTTTCTGAGAATGCTTC |
| Sequence-based reagent | Centromeric α-satellite_R | *Quénet and Dalal, 2014* | PCR primers | TGCATTCAACTCACAGAGTTGAACCTTCC |
| Sequence-based reagent | GAPDH_F | *Quénet and Dalal, 2014* | PCR primers | GCGGTTCCGCACATCCCGGTAT |
| Sequence-based reagent | GAPDH_R | *Quénet and Dalal, 2014* | PCR primers | CCCCACGTCGCAGCTTGCCTA |

*Continued on next page*

*Continued*

| Reagent type (species) or resource | Designation | Source or reference | Identifiers | Additional information |
|---|---|---|---|---|
| Peptide, recombinant protein | CENP-A/H4 tetramer | EpiCypher | Cat. #: 16–010 | |
| Peptide, recombinant protein | H2A/H2B dimer | EpiCypher | Cat. #: 15–0311 | |
| Peptide, recombinant protein | H3/H4 tetramer | EpiCypher | Cat. #: 16–0008 | |
| Commercial assay or kit | TMR-Block | New England Biolabs | Cat. #: S9106S | |
| Commercial assay or kit | TMR-Star | New England Biolabs | Cat. #: S9105S | |
| Commercial assay or kit | NeonTM Transfection System 100 µL kit | ThermoFisher Scientific | Cat. #: MPK10025 | |
| Software, algorithm | R | https://www.r-project.org/ | RRID:SCR_002865 | |
| Software, algorithm | Gwyddion | http://gwyddion.net/ | RRID: SCR_015583 | |
| Software, algorithm | NIH ImageJ | https://imagej.net/software/fiji/ | RRID: SCR_003070 | |
| Software, algorithm | ggplot2 | https://cran.r-project.org/web/packages/ggplot2/index.html | RRID: SCR_014601 | |
| Software, algorithm | CRaQ | http://facilities.igc.gulbenkian.pt/microscopy/microscopy-macros.php | | |
| Software, algorithm | Adobe Photostop | https://www.adobe.com/products/photoshop.html | RRID: SCR_014199 | |
| Software, algorithm | MATLAB | https://www.mathworks.com/products/matlab.html | RRID: SCR_001622 | |
| Software, algorithm | MatLabTrack | https://sourceforge.net/projects/single-molecule-tracking/ | | |
| Other | Vectashield with DAPI | Vector Laboratories | H-1200 | Stain nuclei |

## In vitro reconstitution

In vitro reconstitution of CENP-A (CENP-A/H4 cat#16–010 and H2A/H2B cat#15–0311, EpiCypher, Research Triangle Park, NC) and H3 (H3/H4 cat#16–0008 and H2A/H2B cat#15–0311, EpiCypher Research Triangle Park, NC) nucleosomes were in vitro reconstituted on either 3.5 kbp plasmid containing four copies of α-satellite DNA or a 8.5 kbp plasmid containing 3 copies of PCAT2 lncRNA gene performed as previously described (*Dimitriadis et al., 2010*; *Walkiewicz et al., 2014*). For quality purposes, an aliquot of each sample was imaged by AFM in non-contact tapping mode, before moving on to high-speed AFM.

## High-speed AFM

In vitro reconstituted CENP-A and H3 chromatin with the addition of CENP-C[CD] (*Melters et al., 2019*) or H1.5 (*Melters and Dalal, 2021*), respectively was imaged with the Cypher VRS (Oxford Instruments, Asylum Research, Santa Barbara, CA) using ultra-small silicon cantilevers (BL-AC10DS with nominal resonances of ~1500 kHz, stiffness of ~0.1 N/m) in non-contact-mode. The V1-grade mica on top of the scanning pilar was peeled and functionalized with 167 nM APS, before 10 µL of sample was added. The sample was incubated for 15–20 minutes before initializing scanning to obtain a density of ~400 nucleosome/µm$^2$. The sample was imaged at a speed of 268.82 Hz (frame rate = 1 Hz or one frame per second) and a resolution of 512x256 points and lines for an area of 400x400 nm (CENP-A and CENP-A +CENP C[CD] samples with a nucleosome density of ~300 nucleosomes/µm$^2$, respectively), 300x300 nm (H3 +H1.5 sample with a nucleosome density of ~200 nucleosomes/µm$^2$),

and 250x250 nm (H3 sample with a nucleosome density of ~200 nucleosomes/$\mu m^2$). As the AFM tip moves from the top of the scan area to the bottom, before moving back up again. This cycle repeats itself. This means that the relative tip location on the scan area between successive images do not perfectly correspond in time. To avoid getting inaccurate tracking data, we limited our analysis to only every other scanned image to guarantee that the relative tip position between two successive analyzed images. Videos were saved in mp4 format, converted to TIFF sequences using Photoshop (Adobe), prepared for single molecule tracking in ImageJ (Fuji), and tracked with MATLAB's Matlab-Track v6.0 package. Obtained nucleosome tracks that were shorter than 10 frames and had single steps exceeding 24 nm or nucleosome tracks with average R-step less than 1 nm and a maximum R-range less than 8 nm were excluded from the analysis. Using ggplot2 package in R (version 4.2.1), drift was visualized and calculated for each individual video and any observed drift was corrected and verified. The remaining nucleosome tracks were subsequently analyzed to obtain the mean square displacement curves, diffusion constant, angle between successive frames, single frame step sizes, the maximum R-step, and R-step range. The R-step is the single-frame displacement in the plane. It is defined as the square root of the sum of the squares of the displacement in the x and y direction [R-step = $\sqrt{(\Delta x^2 + \Delta y^2)}$ ]. The results were visualized using the ggplot2 package in R (version 4.2.1). The diffusion constant for each nucleosome sample was estimated from the initial two points of each MSD (*Vestergaard et al., 2014*). The mean and standard error of the mean are reported as the diffusion constant and uncertainty for each nucleosome sample. The mean diffusion constant is identical to the diffusion constant derived from the first two points of the average MSD curves for each nucleosome sample. To test whether the smaller $D_1$ diffusion constant is indistinguishable from 'stuck' or rejected particle trajectories, we compared the x-axis and y-axis diffusion constants of the 'stuck' particle trajectories to the $D_1$ diffusion constant. We fitted the rejected steps in the x- and y- directions to a single Gaussian distribution and obtained the diffusion constant from the standard deviation of the fitted Gaussian. The effective diffusion constants of the 'stuck' trajectories were compared to the smaller $D_1$ diffusion constant. We used the F-test to compare the Gaussian variances to determine the probability that the smaller $D_1$ diffusion constant is consistent with the 'stuck' particle trajectories. The diffusion constants and step analyses were performed in a double-blind manner.

## Native chromatin immunoprecipitation and western blotting

HeLa cells (CCL-2 from AATC; tested negative for mycoplasma contamination) were grown in DMEM (Invitrogen/ThermoFisher Cat #11965) supplemented with 10% FBS and 1 X penicillin and streptomycin cocktail. nChIP experiments were performed without fixation. After cells were grown to ~80% confluency, they were harvested as described (*Bui et al., 2017*; *Bui et al., 2012*). For best results, the pellet obtained for chromatin was spun-down during the nuclei extraction protocol (*Walkiewicz et al., 2014*) and was broken up with a single gentle tap. Nuclei were digested for 6 minutes with 0.25 U MNase/mL (Sigma-Aldrich cat #N3755-500UN) and supplemented with 1.5 mM $CaCl_2$. Following quenching (10 mM EGTA), nuclei pellets were spun down, and chromatin was extracted gently, overnight in an end-over-end rotator, in low salt solution (0.5 x PBS; 0.1 mM EGTA; protease inhibitor cocktail (Roche cat #05056489001)). nChIP chromatin bound to Protein G Sepharose beads (GE Healthcare cat #17-0618-02) were gently washed twice with ice cold 0.5 x PBS and spun down for 1 min at 4 °C at 800 rpm. Following the first nChIP, the unbound fraction was used for the sequential nChIP. Western analyses were done using LiCor's Odyssey CLx scanner and Image Studio v2.0.

## Quantitative PCR

α-satellite expression levels in HeLa cells that were either mock transfected or transfected GFP-CENP-C (generous gift from Stephan Diekmann) using the Neon Transfection System 100 μL kit (Cat. #: MPK10025, Thermo Fisher Scientific, Waltham, MA) per instructions (*McNulty et al., 2017*; *Quénet and Dalal, 2014*). RNA was extracted, quantified by UV-spectroscopy, and equal quantities were retro-transcribed using Superscript III First-Strand Synthesis kit as described above. Complementary DNA (cDNA) samples were prepared using the iQ SYBR Green supermix (#170–8880; Biorad) following manufacturer's protocol. Control reactions without cDNA were performed to rule out non-specific amplification. The quantitative PCR was run on 'Step one plus Real time PCR' system (Applied Biosystem, Grand Island, NY). Primer sequences are:

The comparative cycle threshold ($C_T$) method was used to analyze the expression level of α-satellite transcripts. $C_T$ values were normalized against the average $C_T$ value of the housekeeping gene GAPDH. Relative fold differences ($2^{-\Delta\Delta C_T}$) are indicated in *Figure 5C*.

## Quench pulse-chase immunofluorescence

To quantify de novo assembled CENP-A particles, we transfected HeLa cells with SNAP-tagged CENP-A under a CMV promoter in combination with either empty vector or GFP-CENP-C (generous gift from Stephan Diekmann) using the Neon Transfection System 100 µL kit (Cat. #: MPK10025, Thermo Fisher Scientific, Waltham, MA) per instructions. The quench pulse-chase experiment was performed according to *Bodor et al., 2012*. In short, following transfection, cells were synchronized with double thymidine block. At the first release TMR-block (S9106S, New England Biolabs, Ipswich, MA) was added per manufactures instruction and incubated for 30 min at 37 °C, followed by three washes with cell culture media. At the second release TMR-Star (S9105S, New England Biolabs, Ipswich, MA) was added per manufactures instructions and incubated for 30 min at 37 °C, followed by three washes with cell culture media. Fourteen hours after adding TMR-Star, cells were fixed with 1% paraformaldehyde in PEM (80 mM K-PIPES pH 6.8, 5 mM EGTA pH 7.0, 2 mM MgCl$_2$) for 10 min at RT. Next, cells were washed three times with ice cold PEM. To extract soluble proteins, cells were incubated with 0.5% Triton-X in CSK (10 mM K-PIPES pH 6.8, 100 mM NaCl, 300 mM sucrose, 3 mM MgCl$_2$, 1 mM EGTA) for 5 min at 4 °C. The cells were rinsed with PEM and fixed for a second time with 4% PFA in PEM for 20 min at 4 °C. Next, the cells were washed three times with PEM. Next, the cells were incubated in blocking solution (1 X PBS, 3% BSA, 5% normal goat serum) for 1 hr at RT. CENP-A antibody (ab13979 1:1000) was added for 1 hr at RT, followed by three washes with 1 X PBS-T and a 10 min incubation with blocking solution at RT. Anti-mouse secondary (Alexa-488 1:1000) was added for 1 hr at RT, followed by three 1 X PBS-T and two 1 X PBS washes. Following air-drying, cells were mounted with Vectashield with DAPI (H-1200, Vector Laboratories, Burlingame, CA) and the coverslips were sealed with nail polish. Images were collected using a DeltaVision RT system fitted with a CoolSnap charged-coupled device camera and mounted on an Olympus IX70. Deconvolved IF images were processed using ImageJ. From up to 22 nuclei, colocalizing CENP-A and TMR-Star foci signal were collected, as well as directly neighboring regions using the CRaQ ImageJ macro (*Bodor et al., 2012*). Background signal intensity was subtracted from corresponding CENP-A and TMR-Star signal intensity before the ratio CENP-A/TMR-Star was determined using the CRaQ ImageJ macro (*Bodor et al., 2012*). Samples were imaged in a double-blind manner. Graphs were prepared using the ggplot2 package for R (version 4.2.1).

## Quantification and statistical analyses

Significant differences for western blot quantification and nucleosome track measurements from HS-AFM analyses were performed using either paired or two-sided t-test, F-test, or one-way ANOVA as described in the Figure legends. Significance was determined at $p < 0.05$.

# Acknowledgements

We thank Drs. Tom Misteli and Sam John, and members of the CSEM laboratory for critical comments and suggestions. We thank Drs. Ankita Saha and Craig Mizzen (deceased, University of Illinois, Urbana-Champaign) for the gift of recombinant H1.5 protein and Stephan Diekmann for the gift of GFP-CENP-C plasmid. We thank Dr. Will Heinz (NCI, CCR, OMAL) for kindly letting us use his custom set-up for HS-AFM and Dr. David Ball (NCI, CCR, OMC) for kindly providing help using MATLAB. We thank the reviewers for very useful feedback. This work was supported by the intramural research fund of the Center for Cancer Research at the National Cancer Institute/NIH (D.P.M., R.S.B., T.R., and Y.D.) and National Heart, Lung, and Blood Institute (K.C.N.).

# Additional information

### Competing interests

Yamini Dalal: Reviewing editor, eLife. The other authors declare that no competing interests exist.

## Funding

| Funder | Grant reference number | Author |
|---|---|---|
| National Institutes of Health | Intramural Research | Daniël P Melters |

The funders had no role in study design, data collection and interpretation, or the decision to submit the work for publication.

## Author contributions

Daniël P Melters, Conceptualization, Data curation, Formal analysis, Validation, Investigation, Visualization, Methodology, Writing – original draft, Writing – review and editing; Keir C Neuman, Conceptualization, Formal analysis, Validation, Investigation, Methodology, Writing – review and editing; Reda S Bentahar, Data curation, Validation, Investigation; Tatini Rakshit, Conceptualization; Yamini Dalal, Conceptualization, Supervision, Writing – original draft, Project administration, Writing – review and editing

## Author ORCIDs

Daniël P Melters https://orcid.org/0000-0003-4809-0562
Keir C Neuman http://orcid.org/0000-0002-0863-5671
Yamini Dalal http://orcid.org/0000-0002-7655-6182

## Decision letter and Author response

Decision letter https://doi.org/10.7554/eLife.86709.sa1
Author response https://doi.org/10.7554/eLife.86709.sa2

## Additional files

### Supplementary files

• MDAR checklist

### Data availability

All data generated and analyzed during this study are included in the manuscript and supporting files; Source data files have been provided for Figures 1-6 and accompanied Figure 1-6 - figure supplements. Entire western blot shown in Figure 1 - figure supplement 1 - source data 1 and Figure 5 - figure supplement 1 - source data 1 are included.

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
