## [Editor Report]

This is an interesting paper that describes the validation of high speed AFM as a tool for measuring the dynamics of individual nucleosomes in vitro. After validating the methodology the authors go on to apply the method to nucleosomes containing the centromere-specific histone variant CENP-A, and they show that addition of the CENP-A binding factor CENP-C radically alters the mobility of centromeric nucleosomes.

---

## [Decision Letter]

**Decision letter after peer review:**

[Editors’ note: the authors submitted for reconsideration following the decision after peer review. What follows is the decision letter after the first round of review.]

Thank you for submitting the paper "Mobility of nucleosomes is regulated by their binding partners" for consideration by *eLife*. Your article has been reviewed by 2 peer reviewers, one of whom is a member of our Board of Reviewing Editors, and the evaluation has been overseen by a Senior Editor. The reviewers have opted to remain anonymous.

While the reviewers agreed that the work was potentially interesting they had significant concerns with the conclusions of the manuscript. We are sorry to say that, after consultation with the reviewers, we have decided that this work will not be considered further for publication by *eLife*.

Specifically, overall the study seems somewhat incremental over previously published work. The reviewers felt that characterization of the nucleosome arrays and what is happening to them on the stage was lacking. Do the DNA binding proteins affect interactions of nucleosomes with the stage? It is difficult to interpret the results without this information. In addition, the H1 binding experiments were poorly controlled and should not be included in the manuscript.

*Reviewer #1 (Recommendations for the authors):*

1. I applaud the authors for applying a novel technique to address the fundamental issue of nucleosome mobility. However, this is a relatively short study that is somewhat incremental from previously published work. For example: "Previously we published HSAFM movies of H3 chromatin and H3 chromatin with linker histone variant H1.5 (Melters and Dalal, 2021). Similar as described above, we tracked individual H3 nucleosomes in real-time without (Figure 2A, Supplemental movie 3, Source data 1) and with H1.5 Figure 2B;" Also see: "Previously we showed that overexpressing CENP-C in HeLa cells resulted in increased clustering of centromeric chromatin and loss of centromeric RNAP2 levels (Melters et al., 2019)…To examine this facet of CENP-C: CENP-A homeostasis, we overexpressed CENP-C to assess if the centromeric transcription is altered" "First, we recapitulated the effects of CENP-C overexpression on the levels of RNAP2 on CENPA chromatin" It appears that some of the data in the manuscript are a repeat of published work or at most an incremental advance.

2. The histone H1 experiments should be removed from the manuscript as they are poorly controlled. There is a long history of reported artifactual and preternatural effects of adding H1 to DNA and nucleosomes. Without evidence that H1 is binding the DNA properly and forming chromatosomes the data cannot be interpreted. Moreover, the fact that the H1 to nucleosome ratio used is 0.2:1 suggest that at most only 1 in 5 nucleosomes could be properly bound by H1and measuring the movement of the entire population is not justified. As noted by the authors the data is very heterogeneous suggesting multiple forms of nucleosomes may be present. And again even this data may be a repeat or incremental (see comment 1).

Overall the manuscript would be improved by focusing on CENP-C and CENP-A nucleosomes for which the most data is available and not make sweeping conclusions about all nucleosome binding proteins (see title).

*Reviewer #2 (Recommendations for the authors):*

1) If the authors want to discuss nucleosome accessibility based on nucleosome mobility, they need to perform additional experiments that would bring explicit data on nucleosome accessibility.

2) To rule out the possibility that reduced mobility upon CENP-CCD or H1.5 addition may be due to the increased interaction between the protein complexes and the surface, other proteins that are known to have no apparent interaction with chromatin (for example, BSA), or proteins that are known to increase chromatin mobility, should be also tested using HS-AFM as controls. Furthermore, to link the in vitro observation to in vivo results, CENP-Cs with the same construct should be used.

3) The title can be more informative. Since this study is basically on CENP-C and CENP-A nucleosomes, the authors could try to include these key words in the title too.

4) The authors should provide a description (or maybe a figure) of the structural domains of CENP-C in the introduction.

5) Figure 2C: The standard error bars of MSD for H3 and H3+H1.5 overlap, which does not indicate a statistically significant difference. The diffusion constants also show no statistically dominant difference. Therefore, we cannot assume that the conclusion "linker H1.5 protein restricts H3 nucleosome mobility" is entirely supported by experimental data.

6) The authors should include in the manuscript gel pictures of the reconstituted polynucleosomes used in this study to show the quality of the reconstitution.

7) It is unclear how much CENP-CCD and H1.5 are bound to polynucleosomes. The authors should report the percentage of bound CENP-CCD and H1.5, for example by performing EMSA and then gel quantification. The authors could include a section where they compare the structures of the polynucleosomes with CENP-CCD or H1.5 compared to polynucleosomes without CENP-CCD or H1.5. Is there a difference in the compaction of the structures?

[Editors’ note: further revisions were suggested prior to acceptance, as described below.]

Thank you for resubmitting your work entitled "Single Molecule Analysis of CENP-A Chromatin by High-Speed Atomic Force Microscopy" for further consideration by *eLife*. Your revised article has been evaluated by Kevin Struhl (Senior Editor) and a Reviewing Editor.

The manuscript has been improved but there are some remaining issues that need to be addressed, as outlined below:

Essential revision:

1) please comment on whether the use of α satellite repeat DNA influences the results as opposed to other DNA with no nucleosome positioning preference.

2) What causes the two Gaussian distributions of nucleosomes? This is a potentially interesting result that is not pursued or explained.

3) The authors hint that they are showing potential functions of CENP-C beyond forming kinetochores by over-expressing CENP-C. Are these physiologically relevant or over-expression artifacts?

4) In the authors' description of the experimental workflow they mention that each movie is corrected for drift and they refer readers to Supplemental Figure 2. I would suggest that the legend of supplemental figure 2 be revised to include a brief description of how the drift correction is done, and perhaps add a bit more detail regarding this correction to the Methods section. Also, on page 12 the authors note that "… with Tween-20 displayed the most amount of drift". When the authors introduce the concept of correcting for drift I assumed that they were referring to instrument drift, but I am not certain how to instrument drift would be impacted by Tween-20. Can the authors please clarify?

5) Page 9: "We reasoned that if the AFM tip altered samples during scanning, it would create a distinctive signature in single particle trajectories." For the non-AFM audience, could the authors provide just half a sentence explaining what the data might have looked like if there were systematic issues with the tip altering the sample?

6) The observation of two D1 values (based on the sum of two Gaussian fits) is interesting. Pausing during diffusing seems like a reasonable explanation. I have two questions related to this point. First, is it possible to collect reaction trajectories that are long enough to conclusively say whether any given nucleosome transitions between D1 and D2? Second, is it possible to do a simple computational simulation of 1D diffusion and include pausing within the simulation to verify that the data would yield an apparent sum of two D values?

7) Page 14: The authors state that there is a 4-fold reduction in the diffusion constant with the addition of CENP-C(CD). I think the value is 2.9-fold.

8) How much is CENP-C over-expressed compared to normal expression levels?

9) Have nucleosome diffusion coefficients been reported from smFRET measurements, and if so, how do they compare to the values reported here?

10) Regarding the diffusion measurements themselves: If a nucleosome moves in 1D relative to a DNA strand then the authors "score" this behavior as 1D diffusion and the data is used towards calculating a diffusion coefficient. But, if the DNA moves relative to the nucleosome, then this would not be "scored" as diffusion because it could not be observed in the measurements, correct? If this is true, do the D1 values actually represent apparent values that reflect a lower bound for the actual diffusion coefficients?

11) Nucleosome tracking: Since this paper has a heavy emphasis on methodology, further details on tracking will be helpful. Since the surface is somewhat crowded and the DNA strings are not always clearly observed (based on supplementary movies), how do the authors know if the tracked structure is histone or nucleosome (and within that classification – mono, di, or multi nucleosome)? It is unclear if the quality of the images allows for these distinctions, and nucleosomes in each of these sub-populations are expected to have different dynamics. Did the authors use height or a feature from the surface topography as a criterion for selecting particles for analysis?

(b) It would be helpful to have higher-resolution images of a dilute CENP-A sample acquired at a lower speed than the current 2s/frame to show the composition and features of the nucleosome arrays at higher resolution. It is unclear from the air AFM images shown in Supplement Figure 1B if the sample is largely individual nucleosome strings or a cluster of strings.

12) Please induce height scale with all images in the paper

13) Experiments with CENP-A and CENP-C: The images in Figure 4B are difficult to interpret as the structures do not look homogeneous. Are these nucleosome clusters of different sizes (if yes, how are nucleosomes tracked in these samples?) Again, a higher-resolution image of the sample at a low scan rate will clarify the composition of the sample. How do we know if the changes to dynamics are not due to changes in the surface from the non-specific binding of the CENP-C fragment? One or two controls would be helpful here: (i) a dose-dependent effect of CENP-A nucleosome dynamics at 2-3 different CENP-C concentrations and/or (ii) the addition of a CENP-C mutant that does not bind CENP-A

14) Nucleosome mobility: The authors imply in several places that the observed dynamics have contributions from nucleosome sliding on DNA. For example, in the abstract, the authors state: "CENP-C reduces… along the chromatin fiber"; Pg. 6 "CENP-C, …directly impacts nucleosome mobility and, surprisingly, also chromatin fiber motion in vitro"; Pg. 18 "These findings suggest that physical properties of DNA dictate mobility along DNA"). In response to reviewers, the authors state, "it is tantalizing to hypothesize that the two Gaussian fittings reflect both 1D sliding and 3D chromatin motion". Without direct evidence, the dual Gaussian can arise from any number of reasons (such as collisions, surface adhesion, etc.) and is heavily influenced by the surface.

15) Comment on biological aspects and implications of the study: While the authors indicate that they have reduced the focus on biological implications, a significant section of the paper is still dedicated to it (the final three figures and a large fraction of the discussion). Connecting the mobility of nucleosomes on a mica surface to chromatin accessibility is a big leap (example in the abstract: "changes which alter chromatin accessibility in vitro…."). This is because it is not established from these data how much of the mobility is due to nucleosomes diffusing on the surface (which is heavily influenced by surface properties and buffer conditions, as the authors clearly demonstrate) versus nucleosomes sliding on DNA and altering the access of proteins to different DNA regions. Without direct observation of nucleosome sliding on DNA (which I think is future work and not reasonable to expect here), the biological implications of observed mobility changes in vitro are highly speculative and should be stated as such.

16) The statement that "excess CENP-C suppresses centromeric chromatin accessibility and transcription in vivo" is not well supported by the data presented in this paper. ChIP reports on chromatin occupancy, but not directly on changes in chromatin accessibility and transcription. Please modify the statement.

147 For the western blot in Supplementary Figure 12, there are no loading controls. Can the authors please perform the western blot again and this time include loading controls?

18) Can you please include information on how the error bar was calculated in Figure 2D, Figure 3B, and Figure 4F?

19) Drift correction: were any spots excluded from a correction? In supplementary fig-2, a few spots appear unchanged (x,y ~ 400,30; 10,300; 180,300 etc). These may simply be how the data shifts post drift correction, but if some data was excluded from correction, please include more methodological details.

---

## [Author Response]

[Editors’ note: the authors resubmitted a revised version of the paper for consideration. What follows is the authors’ response to the first round of review.]

Reviewer #1 (Recommendations for the authors):1. I applaud the authors for applying a novel technique to address the fundamental issue of nucleosome mobility. However, this is a relatively short study that is somewhat incremental from previously published work. For example: "Previously we published HSAFM movies of H3 chromatin and H3 chromatin with linker histone variant H1.5 (Melters and Dalal, 2021). Similar as described above, we tracked individual H3 nucleosomes in real-time without (Figure 2A, Supplemental movie 3, Source data 1) and with H1.5 Figure 2B;" Also see: "Previously we showed that overexpressing CENP-C in HeLa cells resulted in increased clustering of centromeric chromatin and loss of centromeric RNAP2 levels (Melters et al., 2019)…To examine this facet of CENP-C: CENP-A homeostasis, we overexpressed CENP-C to assess if the centromeric transcription is altered" "First, we recapitulated the effects of CENP-C overexpression on the levels of RNAP2 on CENPA chromatin" It appears that some of the data in the manuscript are a repeat of published work or at most an incremental advance.

We thank the reviewer for appreciating the application of HS-AFM in studying nucleosome mobility. We also thank the reviewer for helping us refocus our attention in this manuscript. We refocused our attention on performing and optimizing the many control experiments required to establish hsAFM as a robust quotative tool to measure nucleosome dynamics. To end, we also improved the implementation of the single particle tracking software and extended our downstream analyses.

2. The histone H1 experiments should be removed from the manuscript as they are poorly controlled. There is a long history of reported artifactual and preternatural effects of adding H1 to DNA and nucleosomes. Without evidence that H1 is binding the DNA properly and forming chromatosomes the data cannot be interpreted. Moreover, the fact that the H1 to nucleosome ratio used is 0.2:1 suggest that at most only 1 in 5 nucleosomes could be properly bound by H1and measuring the movement of the entire population is not justified. As noted by the authors the data is very heterogeneous suggesting multiple forms of nucleosomes may be present. And again even this data may be a repeat or incremental (see comment 1).

We thank the reviewer for noting this limitation and we agree with the reviewer. Therefore, we have redirected our focus away from describing H3 and H3+H1.5 as an example of how chromatin compaction factors impact nucleosome mobility and limit our analysis of the H3 and H3+H1.5 data to demonstrate bias in nucleosome tracking data (Supplemental Figures S7 and S8). In addition, we show that mononucleosome and chromatin fiber single-frame step distributions are distinct (see Supplemental Figures S7 and S8). More details about the updated manuscript can be found above.

Overall the manuscript would be improved by focusing on CENP-C and CENP-A nucleosomes for which the most data is available and not make sweeping conclusions about all nucleosome binding proteins (see title).

We thank the reviewer for this suggestion and indeed, we have significantly revised the manuscript to not only focus on CENP-A nucleosomes but especially focus on the technical aspects of using HS-AFM in imaging chromatin by performing various controls as described above in detail.

Reviewer #2 (Recommendations for the authors):1) If the authors want to discuss nucleosome accessibility based on nucleosome mobility, they need to perform additional experiments that would bring explicit data on nucleosome accessibility.

We thank the reviewer for this important suggestion, and we can confirm that we are following up on this. A detailed analysis of nucleosome accessibility will be explored in a followup manuscript. For this manuscript, we refocused to the establishment of HS-AFM as a tool to study nucleosome mobility, as described above.

2) To rule out the possibility that reduced mobility upon CENP-CCD or H1.5 addition may be due to the increased interaction between the protein complexes and the surface, other proteins that are known to have no apparent interaction with chromatin (for example, BSA), or proteins that are known to increase chromatin mobility, should be also tested using HS-AFM as controls. Furthermore, to link the in vitro observation to in vivo results, CENP-Cs with the same construct should be used.

We thank the reviewer for this very insightful comment. We have expanded our study to address various conditions that could impact chromatin mobility and dynamics, ranging from salt concentration, APS concentrations, and Tween-20, as described above in detail.

3) The title can be more informative. Since this study is basically on CENP-C and CENP-A nucleosomes, the authors could try to include these key words in the title too.

We agree that the title needed to be more informative. Therefore, we have not only updated the manuscript, but we have also updated the title to reflect the theme of the manuscript more accurately.

4) The authors should provide a description (or maybe a figure) of the structural domains of CENP-C in the introduction.

We thank the reviewer for suggestion. We have altered the focus of the manuscript away from focusing on the effects of CENP-C on CENP-A chromatin to stringently testing CENP-A chromatin under various conditions by HS-AFM. We therefore thought it would not be in line with the current flow of the manuscript to describe the structural domains of CENP-C extensively in the introduction.

5) Figure 2C: The standard error bars of MSD for H3 and H3+H1.5 overlap, which does not indicate a statistically significant difference. The diffusion constants also show no statistically dominant difference. Therefore, we cannot assume that the conclusion "linker H1.5 protein restricts H3 nucleosome mobility" is entirely supported by experimental data.

We have redone the analysis for both the CENP-A, CENP-A+CENP-C^CD, H3, and H3+H1.5 based on more tracked particles. These extended data led us to reanalyze all data and revise our interpretations accordingly. We no longer make global statements but restrict our conclusions to the data presented in this manuscript.

6) The authors should include in the manuscript gel pictures of the reconstituted polynucleosomes used in this study to show the quality of the reconstitution.

We thank the reviewer for this suggestion. We added supplemental Figure S1 which has an image of a gel showing the purity of histones; an AFM image showing in vitro reconstituted nucleosomes confirming reconstituted polynucleosomes; a MNase ladder showing digested chromatin confirming reconstituted polynucleosomes; and AFM nucleosome measurements confirming the expected nucleosomal dimensions. All these data confirm the quality of the chromatin reconstitution.

7) It is unclear how much CENP-CCD and H1.5 are bound to polynucleosomes. The authors should report the percentage of bound CENP-CCD and H1.5, for example by performing EMSA and then gel quantification. The authors could include a section where they compare the structures of the polynucleosomes with CENP-CCD or H1.5 compared to polynucleosomes without CENP-CCD or H1.5. Is there a difference in the compaction of the structures?

We thank the reviewer for this question. Previous work has established how much CENP-C^CD binds to CENP-A nucleosomes. Specifically, in Falk et al. 2015, Falk et al. 2016, Guo et al. 2017, Melters et al. 2019, and Ali-Ahmed et al. 2019. We used the same CENP-A nucleosome to CENP-C^CD ratio of 1: 2.2 as established and subsequently used in these studies.

We are very excited to study how H1.5 might or might not interact with CENP-A chromatin. Indeed, this is an ongoing study in the lab that we will publish in a manuscript that is preparation. In short, we see CENP-A chromatin compaction by H1.5. We are working out the details how this mechanistically happens and if there is competition between CENP-C^CD and H1.5.

[Editors’ note: further revisions were suggested prior to acceptance, as described below.]

The manuscript has been improved but there are some remaining issues that need to be addressed, as outlined below:Essential revision:1) Please comment on whether the use of α satellite repeat DNA influences the results as opposed to other DNA with no nucleosome positioning preference.

We thank the reviewer for this important question. We are aware that DNA sequence can influence nucleosome dynamics, including nucleosome positioning. Indeed, α-satellite DNA can position nucleosomes well, leading it to be used in the now-iconic crystal structure solved by Luger et al. In our study we used a 3.5 kbp plasmid contains 4 copies of a-satellite DNA, which make up less than 20% of the plasmid. The majority of the plasmid lacks nucleosome positioning sequences. Therefore, we accepted the reviewer’s suggestion to expand our study to also include a plasmid that lacks any known nucleosome positioning sequence. We used a 8.5 kbp plasmid that contains a copy of the PCAT2 gene, which is a lncRNA gene near the MYC gene on the human 8q24 locus. In cancers, CENP-A can go to ectopic loci, and the PCAT2 locus is one such known ectopic sites (Athwal et al. 2015, Nye et al. 2018, Ganesan et al. 2022). When we compared the MSD curves and diffusion constants of CENP-A nucleosomes reconstituted on the PCAT2 plasmid to other controls conditions, we observed that CENP-A nucleosomes have a similar MSD curve and diffusion constant as the high salt control condition (updated Figure 3 and Figure 3 —figure supplement 1-3). We found a higher diffusion constant and MSD curve for CENP-A nucleosome on PCAT2 plasmid versus a-satellite plasmid. One possibility is that DNA sequence indeed can influence nucleosome mobility. Another possibility is that the length difference between the two plasmids influences nucleosome mobility. We will work out these details in follow-up experiments more carefully comparing the effect of various positioning sequences- the senior author of this ms performed her PhD working on this exact issue with Arnie Stein and Minou Bina in the 2000s; so it is certainly of deep interest to the Dalal lab.

2) What causes the two Gaussian distributions of nucleosomes? This is a potentially interesting result that is not pursued or explained.

We agree with the reviewer. The precise cause of the two Gaussian distribution is not fully clear to us. As a first step, we elaborated on this interesting finding by addressing whether the smaller D_1_ diffusion constant could simply arise from nucleosome “sticking” to the mica surface. For this, we compared the diffusion constant of nucleosome trajectories that were rejected for being potentially “stuck” with the smaller D_1_ diffusion constant. Trajectories were rejected if their average R step was less than 1 nm and their maximum R range was less than 8 nm. Next, we fitted the “stuck” steps in the x and y diffusion constant, as described in Figure 2C, D and Figure 2 —figure supplement 1B. The effective diffusion constant of the “stuck” trajectories were compared with the smaller D_1_ diffusion constant. We used the F-test to compare the gaussian variances to determine the probability that the smaller diffusion constant is consistent with the “stuck” nucleosomes. With the exception of the 2x APS and low salt condition, the diffusion constants were significantly different. These results are not consistent with the smaller D_1_ diffusion constant being the same as the “stuck” trajectories (see new Figure 3 —figure supplement 5). We are therefore more confident that the D_1_ and D_2_ Gaussian distributions represent two distinct mobility states. Further work will be necessary to understand the cause of the two states, and is slated for a follow up study.

3) The authors hint that they are showing potential functions of CENP-C beyond forming kinetochores by over-expressing CENP-C. Are these physiologically relevant or over-expression artifacts?

We thank the reviewer for this question. CENP-C overexpression is not a common event in cells. In fact, in tumors it was found that CENP-C expression is not commonly altered, whereas the expression of other centromeric and kinetochore proteins is (Zhang et al. 2016 Nat Comm). In a follow-up study we will elaborate on this in more detail. In a preview, we will show (see Author response image 1) that overexpression of CENP-C leads to extensive mitotic defects that can be rescued by co-expressing with mutant histone variants that function as a sink for excess CENP-C (H3^CpA CTD^) or a mutant histone variant that cannot bind CENP-C (CENP-A^∆CTD^).

**Author response image 1. sa2fig1:** 

4) In the authors' description of the experimental workflow they mention that each movie is corrected for drift and they refer readers to Supplemental Figure 2. I would suggest that the legend of supplemental figure 2 be revised to include a brief description of how the drift correction is done, and perhaps add a bit more detail regarding this correction to the Methods section. Also, on page 12 the authors note that "… with Tween-20 displayed the most amount of drift". When the authors introduce the concept of correcting for drift I assumed that they were referring to instrument drift, but I am not certain how to instrument drift would be impacted by Tween-20. Can the authors please clarify?

We have updated Figure 1 —figure supplement 2 to more clearly show how we corrected for sample drift. In short as shown in Figure 1 —figure supplement 2A, for each movie we visualized the raw tracks. Some of the tracks appear to be immobile. These tracts were rejected for further analysis if they had an average R step smaller than 1nm and a maximum R range of smaller than 8 nm. These immobile tracts display a classic pattern of drift where over time the tract would move in the exact same direction as other immobile tracts. By taking the first and last position of several immobile particles, divided by the number of steps, the correction per step was calculated. Next, we verified the correction by visualizing the steps as we did when we determined the drift. In Figure 1 —figure supplement 2B, we show four examples of drift correction, but please note that this was done for each movie individually.

5) Page 9: "We reasoned that if the AFM tip altered samples during scanning, it would create a distinctive signature in single particle trajectories." For the non-AFM audience, could the authors provide just half a sentence explaining what the data might have looked like if there were systematic issues with the tip altering the sample?

We agree with the reviewer that such a statement would make it more obvious to the non-AFM audience. As such, we have updated the text on page 9.

6) The observation of two D1 values (based on the sum of two Gaussian fits) is interesting. Pausing during diffusing seems like a reasonable explanation. I have two questions related to this point. First, is it possible to collect reaction trajectories that are long enough to conclusively say whether any given nucleosome transitions between D1 and D2? Second, is it possible to do a simple computational simulation of 1D diffusion and include pausing within the simulation to verify that the data would yield an apparent sum of two D values?

We thank the reviewer for these insightful questions! (1) We have collected nucleosome trajectories that show what appears to be transitions from D_1_ to D_2_ states and vice versa. We show a collage of these trajectories in Figure 3 —figure supplement 4A. Additionally, when we looked at the single step size distribution for each trajectory, we observed that there is a very broad range of step sizes, indicating that it is plausible that individual nucleosomes transition between D_1_ and D_2_ states (Figure 3 —figure supplement 4B). (2) Instead of developing and running a simulation, we performed a test to determine if the smaller D_1_ diffusion constant could be the result of “sticking” nucleosomes. In our response to Point 2, we elaborate in detail how we performed this analysis. Finally, we will try to understand the origin of the two mobility populations in a follow-up study.

7) Page 14: The authors state that there is a 4-fold reduction in the diffusion constant with the addition of CENP-C(CD). I think the value is 2.9-fold.

We thank the reviewer for catching this error, which has been corrected.

8) How much is CENP-C over-expressed compared to normal expression levels?

Based on Western blot data (Source Data 6), CENP-C was modestly over-expressed by 2.8-fold from normal. We have updated the text to reflect the level of overexpression.

9) Have nucleosome diffusion coefficients been reported from smFRET measurements, and if so, how do they compare to the values reported here?

As far as we are aware and have found in the literature, smFRET has not been used for determining nucleosome diffusion coefficients. Beat Fierz’s group has done beautiful work using smFRET to study nucleosome array dynamics, but his work has not determined the diffusion coefficients. On the other hands, other imaging-based technologies have been deployed to determine the diffusion coefficient of molecules, such as FCS. We have updated the Discussion section to include the comparison of our results with published work.

10) Regarding the diffusion measurements themselves: If a nucleosome moves in 1D relative to a DNA strand then the authors "score" this behavior as 1D diffusion and the data is used towards calculating a diffusion coefficient. But, if the DNA moves relative to the nucleosome, then this would not be "scored" as diffusion because it could not be observed in the measurements, correct? If this is true, do the D1 values actually represent apparent values that reflect a lower bound for the actual diffusion coefficients?

We thank the reviewer for this important question. Under ideal conditions, we would be able to track nucleosomes along the DNA (1D motion), as well as the chromatin fiber as a whole (2D motion). As the reviewer can appreciate in the movies, DNA frequently moves out of focus (away from the mica surface), making continuous tracking difficult. This is akin to a particle moving out of focus in live cell imaging, or because DNA moves too fast to be captured during scanning process. This latter is aggerated in movies where we scanned at higher resolution, at the cost of scanning speed (see images in response to Comment 11b). There is an upgrade available for the HS-AFM machine that can scan at higher speed with greater resolution, which might provide the technical tools to gain spatiotemporal resolution to observe different diffusion patterns in greater detail.

11) Nucleosome tracking: Since this paper has a heavy emphasis on methodology, further details on tracking will be helpful. Since the surface is somewhat crowded and the DNA strings are not always clearly observed (based on supplementary movies), how do the authors know if the tracked structure is histone or nucleosome (and within that classification – mono, di, or multi nucleosome)? It is unclear if the quality of the images allows for these distinctions, and nucleosomes in each of these sub-populations are expected to have different dynamics. Did the authors use height or a feature from the surface topography as a criterion for selecting particles for analysis?(b) It would be helpful to have higher-resolution images of a dilute CENP-A sample acquired at a lower speed than the current 2s/frame to show the composition and features of the nucleosome arrays at higher resolution. It is unclear from the air AFM images shown in Supplement Figure 1B if the sample is largely individual nucleosome strings or a cluster of strings.

We thank the reviewer for these questions. (a) Particles were identified as nucleosomes if their diameter was 10-12 nm. Histone H2A/H2B dimers and CENP-A/H4 tetramers are much smaller (see Author response image 2) and were not tracked. Post-analysis, we did a spot check to verify the presence of entry and exit DNA strands from the tracked particles, as these were not always visible in all frames as the reviewer accurately pointed out. We have updated the methods section, as well as the Results section to more clearly explain how nucleosomes were identified.

**Author response image 2. sa2fig2:** An example of a failed in vitro reconstitution imaged by conventional in air AFM, where only a few nucleosomes (red arrow) are formed. CENP-A/H4 tetramers (green arrow) can be easily distinguished from H2A/H2B dimers (pink arrows). In all cases, a nucleosome is much larger in diameter and height than either CENP-A/H4 tetramer or H2A/H2B dimers.

(b) We have also imaged CENP-A chromatin at the highest resolution that the Cypher VRS permits, which is 744 x 1024 points & lines. As you can see from the screenshots of HS-AFM Videos (Author response image 3), there is a marginal increase in observed resolution. The mica and sampels were prepared similar as in the main experiment and the sample was imaged using the same parameters, with the only exeption being resolution. The same 400x400 nm area was first scanned at the highest resolution and subsequently at the settings used in the rest of the manuscript. The topological resolution did not improve noticeable under the highest resolution conditions compared to the settings used in the manuscript. Temporal resolution was unfortunatly lost.

**Author response image 3. sa2fig3:** A side-by-side comparison of the same region that was imaged at two different resolutions and consequently, speeds. The left image was scanned at the maximum resolution of 744 x 1024 lines & points at 1 frame per 4 seconds. The right image was scanned at 512 x 256 lines & points at 1 frame per second.

12) Please induce height scale with all images in the paper

We have added height scale bars next to all AFM images.

13) Experiments with CENP-A and CENP-C: The images in Figure 4B are difficult to interpret as the structures do not look homogeneous. Are these nucleosome clusters of different sizes (if yes, how are nucleosomes tracked in these samples?) Again, a higher-resolution image of the sample at a low scan rate will clarify the composition of the sample. How do we know if the changes to dynamics are not due to changes in the surface from the non-specific binding of the CENP-C fragment? One or two controls would be helpful here: (i) a dose-dependent effect of CENP-A nucleosome dynamics at 2-3 different CENP-C concentrations and/or (ii) the addition of a CENP-C mutant that does not bind CENP-A

We thank the reviewer for this suggestion. How we distinguish nucleosomes from CENP-A/H4 tetramers from H2A/H2B dimers, please see response to Point 11. To address whether CENP-C modulates CENP-A nucleosomes in a dose-dependent manner, we performed additional HS-AFM movies of CENP-A chromatin adding 1x and 4x levels of CENP-C^CD^. In other words, we added in total either 1, 2, or 4 CENP-C^CD^ molecules per CENP-A nucleosome (1x, 2x, or 4x CENP-C^CD^). By adding two concentration levels for CENP-C^CD^, we observed that 1x CENP-C^CD^ has no effect on CENP-A nucleosome MSD curves (updated Figure 4D) and the diffusion constant (updated Figure 4E), whereas 4x CENP-C^CD^ MSD curves (updated Figure 4D) and diffusion constant (updated Figure 4E) was very similar to 2x CENP-C^CD^. To further clarify the experimental set-up, we included a graphical representation of the CENP-C protein, which fragment was used (central domain), and at what molar concentration (updated Figure 4A). Furthermore, we updated Figure 3 —figure supplement 1-3 and the text, both in the Results section, discussion, and material and methods. We interpret these results to mean that CENP-C^CD^ regulates CENP-A nucleosome motions not like a rheostat, but instead more like a switch. The consequence of this would be that CENP-A nucleosome motions will only stop once a critical mass of CENP-C^CD^ is bound to CENP-A chromatin.

14) Nucleosome mobility: The authors imply in several places that the observed dynamics have contributions from nucleosome sliding on DNA. For example, in the abstract, the authors state: "CENP-C reduces… along the chromatin fiber"; Pg. 6 "CENP-C, …directly impacts nucleosome mobility and, surprisingly, also chromatin fiber motion in vitro"; Pg. 18 "These findings suggest that physical properties of DNA dictate mobility along DNA"). In response to reviewers, the authors state, "it is tantalizing to hypothesize that the two Gaussian fittings reflect both 1D sliding and 3D chromatin motion". Without direct evidence, the dual Gaussian can arise from any number of reasons (such as collisions, surface adhesion, etc.) and is heavily influenced by the surface.

We agree with the reviewer that various conditions may influence the behavior of nucleosomes and nucleosome arrays. For that reason, we tested several conditions, ranging from salt concentrations, how sticky the mica surface is (APS concentration), and Tween-20 to modify the non-specific hydrophobic interactions. In all tested conditions, which may represent the largest data set of nucleosomes observed by HS-AFM, we observed that the sum of two Gaussians best fitted the single step distribution (see Figure 3D and Figure 3 —figure supplement 2). That said, we don’t fully understand the precise source of the diffusion constants derived from the sum of two Gaussians. Excitingly, a recent paper from our department colleagues in Gordon Hager’s lab (Wagh et al. 2023 Sci Adv) independently performed single particle tracking analysis of H2B-HALO in vivo, also report two distinct mobility states. We have included this study in the discussion. Furthermore, we performed an additional control as described in our response to Point 2. Finally, in a follow-up study, we will delve into the origin of the two Gaussians.

15) Comment on biological aspects and implications of the study: While the authors indicate that they have reduced the focus on biological implications, a significant section of the paper is still dedicated to it (the final three figures and a large fraction of the discussion). Connecting the mobility of nucleosomes on a mica surface to chromatin accessibility is a big leap (example in the abstract: "changes which alter chromatin accessibility in vitro…."). This is because it is not established from these data how much of the mobility is due to nucleosomes diffusing on the surface (which is heavily influenced by surface properties and buffer conditions, as the authors clearly demonstrate) versus nucleosomes sliding on DNA and altering the access of proteins to different DNA regions. Without direct observation of nucleosome sliding on DNA (which I think is future work and not reasonable to expect here), the biological implications of observed mobility changes in vitro are highly speculative and should be stated as such.

We thank the reviewer for pointing out the speculative nature of extrapolating HS-AFM data to in vivo events. We accept the advice to tone down our excitement about what is happening in vivo. We have updated the manuscript accordingly in both the abstract, introduction, results, and discussion.

16) The statement that "excess CENP-C suppresses centromeric chromatin accessibility and transcription in vivo" is not well supported by the data presented in this paper. ChIP reports on chromatin occupancy, but not directly on changes in chromatin accessibility and transcription. Please modify the statement.

We have modified the statement to represent the data more accurately: “Excess CENP-C suppresses centromeric RNAP2 levels and centromeric transcription in vivo".

17) For the western blot in Supplementary Figure 12, there are no loading controls. Can the authors please perform the western blot again and this time include loading controls?

We apologize for this oversight. We have added H2A as a loading control to Figure 5 —figure supplement 1 and updated the Figure 5 - figure supplement 1 - source data 1 as well.

18) Can you please include information on how the error bar was calculated in Figure 2D, Figure 3B, and Figure 4F?

The error bar in Figures 2D, 3B, and 4F represent the standard error, as reported in the methods section. For clarity, we have updated the figure legends.

19) Drift correction: were any spots excluded from a correction? In supplementary fig-2, a few spots appear unchanged (x,y ~ 400,30; 10,300; 180,300 etc). These may simply be how the data shifts post drift correction, but if some data was excluded from correction, please include more methodological details.

We thank the reviewer for this comment and please note our response to Point 4. When drift correction was performed, all data points within that movie were corrected. The amount of correction was determined in a movie-specific manner.